# Human organotypic brain slice culture: a novel framework for environmental research in neuro-oncology

Vidhya M Ravi[1,2,3,8,]*, Kevin Joseph[1,3,8,]*, Julian Wurm[1,8], Simon Behringer[1,8], Nicklas Garrelfs[1,8], Paolo d' Errico[4,8], Yashar Naseri[3,8], Pamela Franco[3,8], Melanie Meyer-Luehmann[4,8], Roman Sankowski[5,8], Mukesch Johannes Shah[3,8], Irina Mader[6], Daniel Delev[7], Marie Follo[8,9], Jürgen Beck[3,8], Oliver Schnell[1,3,8], Ulrich G Hofmann[2,3,8,†], Dieter Henrik Heiland[1,3,8,†]

When it comes to the human brain, models that closely mimic in vivo conditions are lacking. Living neuronal tissue is the closest representation of the in vivo human brain outside of a living person. Here, we present a method that can be used to maintain therapeutically resected healthy neuronal tissue for prolonged periods without any discernible changes in tissue vitality, evidenced by immunohistochemistry, genetic expression, and electrophysiology. This method was then used to assess glioblastoma (GBM) progression in its natural environment by microinjection of patient-derived tumor cells into cultured sections. The result closely resembles the pattern of de novo tumor growth and invasion, drug therapy response, and cytokine environment. Reactive transformation of astrocytes, as an example of the cellular nonmalignant tumor environment, can be accurately simulated with transcriptional differences similar to those of astrocytes isolated from acute GBM specimens. In a nutshell, we present a simple method to study GBM in its physiological environment, from which valuable insights can be gained. This technique can lead to further advancements in neuroscience, neuro-oncology, and pharmacotherapy.

## Introduction

Glioblastoma (GBM) is one of the most malignant of brain tumors (54% of all gliomas and 16% of all primary brain tumors [Stupp et al, 2005]), with an average life expectancy of ≤14 mo postdiagnosis (Ramirez et al, 2013; Urbańska et al, 2014; Williams, 2014; Kalita et al, 2016; Staller, 2016; Batash et al, 2017). One hallmark of GBM is the aggressive infiltration into healthy brain regions (Müller et al, 2014; Xie et al, 2014; Darmanis et al, 2017; Birch et al, 2018). These tumors are exclusive to the central nervous system (CNS), with extracranial metastases being rare (Ray et al, 2015). This aspect is an indicator that the CNS microenvironment is essential for the maintenance and proliferation of GBM. The crosstalk between GBM and its microenvironment is of great interest, and improved experimental models must better investigate this mutual interaction.

Until recently, gliomas have been studied using simple but incomplete models based on 2D monolayer cultures of cell lines derived from primary tumor specimens, where the micromilieu is partially simulated through the use of supplements (Eisemann et al, 2018). The use of 2D monocultures to study the malignant properties of GBM cell lines would therefore result in an inability to study their tissue-specific functions and morphological organization, and they cannot recapitulate every aspect of the tumor microenvironment (TME) (Bissell, 1981). The TME plays a critical role in tumor progression as it controls the rate at which tumors can grow and proliferate (Hanahan & Coussens, 2012). The two key biophysical parameters controlling a tumor's interactions with its microenvironment are molecular gradients and mechanical stresses (Butcher et al, 2009). In a further step, murine models were implemented to study tumor propagation via orthotopic or subcutaneous xenografts of tumor cells (Jung et al, 2018). However, there are two decisive disadvantages with these models in addition to their labor intensiveness: (i) the reported models do not sufficiently simulate the malignant properties of GBM tumors (Jackson & Thomas, 2017) and (ii) the complex microenvironment with its dynamic changes and influences because of cellular

[1]Translational NeuroOncology Research Group, Medical Center, University of Freiburg, Freiburg im Breisgau, Germany   [2]Neuroelectronic Systems, Medical Center, University of Freiburg, Freiburg im Breisgau, Germany   [3]Department of Neurosurgery, Medical Center, University of Freiburg, Freiburg im Breisgau, Germany   [4]Department of Neurology, Medical Centre, University of Freiburg, Freiburg im Breisgau, Germany   [5]Institute of Neuropathology, Medical Center, University of Freiburg, Freiburg im Breisgau, Germany   [6]Clinic for Neuropediatrics and Neurorehabilitation, Epilepsy Center for Children and Adolescents, Schön Klinik, Vogtareuth, Germany   [7]Department of Neurosurgery, University of Aachen, Aachen, Germany   [8]Faculty of Medicine, University of Freiburg, Freiburg im Breisgau, Germany   [9]Department of Medicine I, Medical Center, University of Freiburg, Freiburg im Breisgau, Germany

Correspondence: Vidhya.ravi87@gmail.com; dieter.henrik.heiland@uniklinik-freiburg.de
*Vidhya M Ravi and Kevin Joseph contributed equally to this work.
†Ulrich G Hofmann and Dieter Henrik Heiland contributed equally to this work.

components is insufficiently mapped for interspecies studies because of differences in CNS physiology (Wellbourne-Wood & Chatton, 2018).

This has resulted in efforts to reduce or replace animal models with new advances such as (i) 3D embedded matrices with collagen and hyaluronic acid (Fernandez-Fuente et al, 2014) and (ii) microfilters (Hi-spots) (Biggs et al, 2011). Although these techniques are quick, commercially available, and easy to perform, they have several limitations, such as the stiffness of the collagen matrix, the concentration of hyaluronic acid, and the lack of control over the cellular composition in Hi-spots, which can affect the invasive properties of the glioma cell line being studied (Rao et al, 2013; Grotzer et al, 2016). These assays do not take into account the unique myelinated nerve fibers or the ECM between neuronal and glial processes in the brain parenchyma, which substantially influences the invasive properties of tumors (Cuddapah et al, 2014). This is further emphasized by the fact that infiltrated tissue is characteristically stiffer than normal tissue because of resident fibroblast remodeling and increased contractility of the transformed epithelium (Butcher et al, 2009; Leventa et al, 2009).

To overcome these described limitations, we propose the use of adult human organotypic brain slice cultures. Organotypic slice cultures preserve in vivo morphology and the cellular architecture of neuronal tissue, lasting from a few days up to weeks in culture (Cavaliere et al, 2016). However, most experimental reports presenting this technique have made use of postnatal murine tissue, with limited evidence that tissue obtained from animals at later stages of development can be reliably cultured for extended periods (Noraberg et al, 1999).

Because GBM is a characteristic postadolescent disease (Alexander & Cloughesy, 2017), reliable culture of tissue obtained from age-appropriate neuronal specimens is paramount (Eisemann et al, 2018). Organotypic cultures have been used to

study the invasive properties of tumors by using tumor spheroids seeded in human (Jung et al, 2002) and murine organotypic sections. These findings report overestimated spheroid penetration properties into the surrounding tissue (Eisemann et al, 2018) and show differences between species (Zhang et al, 2016; Kallendrusch et al, 2017). Murine models of GBM are thus not a true reflection of the properties that human GBM exhibit, whereas the use of human GBM cell lines in animals can cause cross-species–specific reactivity that can lead to weak assumptions in GBM progression and response to treatment (Huszthy et al, 2012).

In this work, we cultured human brain sections to study the invasive properties of tumor in its appropriate microenvironment. Patient-derived GBM cells were microinjected into the cultured sections to study the interaction of GBM in a "native-like" environment. We present a robust and reproducible method that has been systematically evaluated using a battery of validation techniques. As an example of the utility of this approach, we further demonstrate that the extraction of specific cell types from the cultured sections is possible, allowing the determination of changes in the mRNA expression profiles while located in their native environment.

## Results

Fresh neocortical tissue samples were obtained from donors undergoing therapeutic resections for either epilepsy (N = 5) or GBM (N = 21) (Fig 1A). In the case of tissue sourced from tumor patients, the average age of the donors was 63 ± 12 yr and that of epileptic donors was 24 ± 20 yr (Fig 1B, patient information is given in Table S1). For this study, the sections were obtained from frontal lobe, parietal lobe, occipital lobe, and temporal lobe (Fig 1C). After preoperative contrast magnetic resonance imaging (MRI)–guided planning,

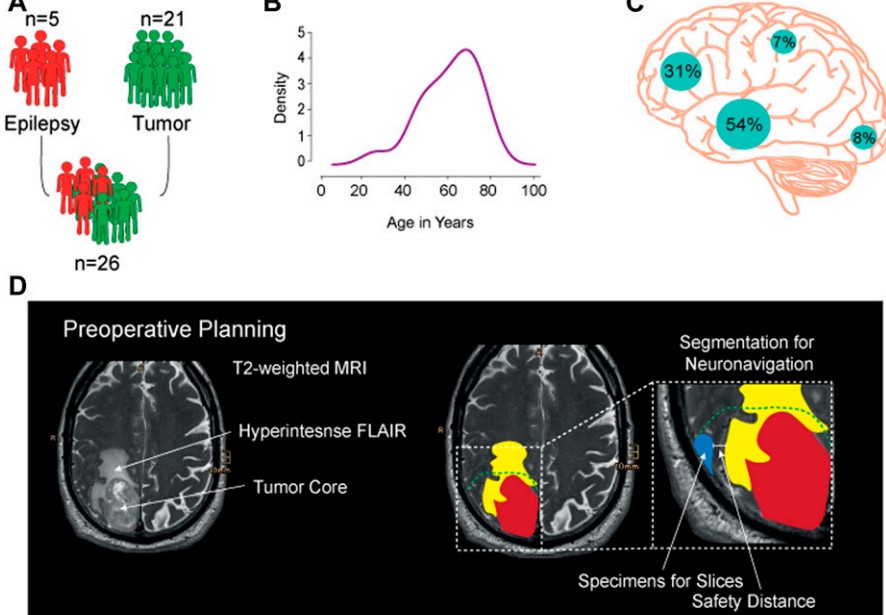

**Figure 1. Validation of tissue collection.**
**(A)** Neural tissue samples from N = 26 patients (n = 5 patients with epilepsy and n = 21 tumor patients) were used in this work. **(B)** Density diagram of the age of tissue donors that contributed to this study. **(C)** Distribution of the anatomical regions that the tissue used in this study was sourced from (frontal lobe: 31%, parietal lobe: 8%, occipital lobe: 8%, and temporal lobe: 54%). **(D)** Preoperative planning carried out before the resection procedure to ascertain the "health" status of the resected access cortex during tumor surgery. There was a safety distance of 2 cm from the infiltrating cortex to avoid contamination by GBM cells.

distant cortex without tumor infiltration (minimum of 2 cm from Flair hyper intensive regions), guided by intraoperative neuro-navigation (Fig 1D), was resected and used in this study. This step is crucial because the tumor-infiltrated tissue does not have the same properties as healthy neuronal tissue, which could potentially affect the integrity of our model.

To validate the infiltration status of each resected tissue sample, hematoxylin and eosin staining was carried out in addition to MRI (Fig 2A). In the case of tissue resected from patients with epilepsy, the tissue was resected as previously described (Rassner et al, 2016). Only the cortical access tissue and not the epileptic focus was made use of in this study.

## Vitality of cultured sections

All experiments were carried out using 300-$\mu$m-thick sections that were obtained using a vibratome (Leica VT1200) and cultured in six-well plates for up to 2 wk. The vitality of the sections was characterized using immunohistochemistry, electrophysiology, ELISA, and RNA-sequencing methods (Fig 2B).

The preservation of the neuronal population in cultured neo-cortical sections was our paramount goal and therefore the

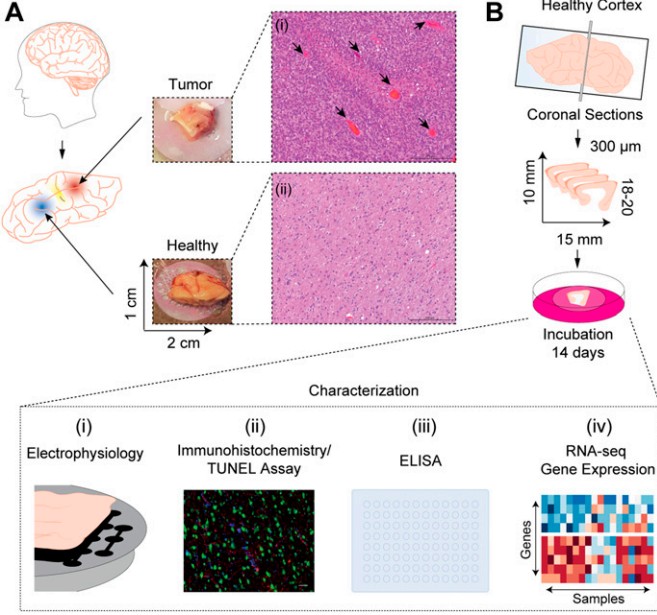

**Figure 2. Illustration of workflow.**
**(A)** The relative separation of the tumor-infiltrated cortex from the healthy cortex was verified by means of hematoxylin and eosin staining. Microvascular proliferation (black arrows) is seen in (i) tumor tissue compared with (ii) healthy cortex. **(B)** Illustration of the workflow for the generation of the sections used for organotypic brain cultures. A vibratome was used to obtain 18–20 coronal sections of 300 $\mu$m thickness from every 1 × 2-cm tissue block, with dimensions of ~10 × 15 mm. Sections were then transferred to the nylon membrane of an insert in a six-well plate using the blunt, fire polished end of a glass Pasteur pipette and incubated at 37°C with the surface of the media contacting the membrane, enabling diffusion for up to 14 d. Sections were then characterized at different time points using (i) MEA electrophysiology to confirm neuronal activity, (ii) immunohistochemistry to evaluate the cell composition and loss over time, (iii) ELISA to evaluate the cellular damage or cytokine measurement, and (iv) MinION RNA-sequencing/qPCR to evaluate profile changes in gene expression.

numbers of neurons in each were quantified (number of NeuN+ cells per unit area). In contrast to the maintenance of astrocytes in culture, previous reports have suggested difficulty in maintaining a viable neuronal population, especially in tissue sourced from adult donors (Humpel, 2015). This is because of axotomy during the process of tissue resection and sectioning. Changes in neuronal localization at different time points were assessed and compared with that of acute brain sections (Fig 3A) from N = 20 patients. We found that the cytoarchitecture of the human brain is retained in the sections grown for up to 7 d in serum-free growth medium (Fig 3A, middle panel). The sections seem to lose defined layering by DIV14. These results are in agreement with previously reported work (Chaichana et al, 2007). Immunohistochemistry analysis shows an expected reduction in the number of neuronal cells in the sections between acute and cultured sections ($P < 0.0001$, one-way ANOVA with Bonferroni correction for multiple comparisons, Fig 3B). However, the rate of change in the number of neurons within the culture was maintained, with no drastic drops detected over the course of the culture period (Fig 3C). Alterations in the activated astrocytes (Glial fibrillary acidic protein [GFAP]) were investigated in brain sections cultured for all the time points studied (Fig 3A). The quantitative analysis of immunohistochemistry results (Fig 3D) shows that there is a significantly higher population of GFAP+ astrocytes immediately postsectioning, with a 21.2% reduction at 1 d postsectioning and plating ($P < 0.00001$, unpaired $t$ test with Welch's correction). This is in agreement with previous reports, accounting for the inflammatory activation of glial cells because of tissue trauma, postresection, and postsectioning (Eisemann et al, 2018). Post DIV1, we report no significant difference over the course of the culture period (DIV4 = −5.97%, DIV7 = −8.15%, DIV14 = 5.58%, one-way ANOVA with Bonferroni correction for multiple comparisons, Fig 3E). To prevent astrocyte differentiation into its inflammatory subtype, the sections were cultured in serum-free medium (Conti et al, 2005).

To further validate the model, two different methods were used to determine the health of the neuronal sections over the course of the entire culture period. The first method quantified DNA fragments within apoptotic cells using TUNEL staining in both acute and cultured brain sections (DIV1–DIV14) with N = 4 patients. Because the cells at the faces of the sections are damaged by the slicing procedure itself, the levels of cell death detected were significantly higher in freshly sectioned tissue (acute) compared with the cultured sections. The total number of cells that were both TUNEL+ and DAPI+ were compared between the different time points. As expected, there is a 22% decrease in TUNEL+ cells in DIV1 compared with immediately postsectioning ($P = 0.0004$, unpaired $t$ test with Welch's correction, Fig 3F), which then stays within 12% of the value measured on DIV1 until DIV14 (DIV4 = −6.75%, DIV7 = −11.35%, DIV14 = 0.61%, one-way ANOVA with Bonferroni correction for multiple comparisons, Fig 3G).

The second assay quantified lactate dehydrogenase (LDH) released from cells with damaged cellular membranes, which is indicative of necrotic/apoptotic cells. Medium taken from sections 4 h postplating versus 1 d in culture shows a 29% higher presence of LDH in the medium immediately postresection, corresponding to the increased number of TUNEL+ cells compared with cultured sections ($P < 0.00001$, unpaired $t$ test with Welch's correction, Fig 3H). In comparison with DIV1, the LDH values for DIV4, 7, and 14 are

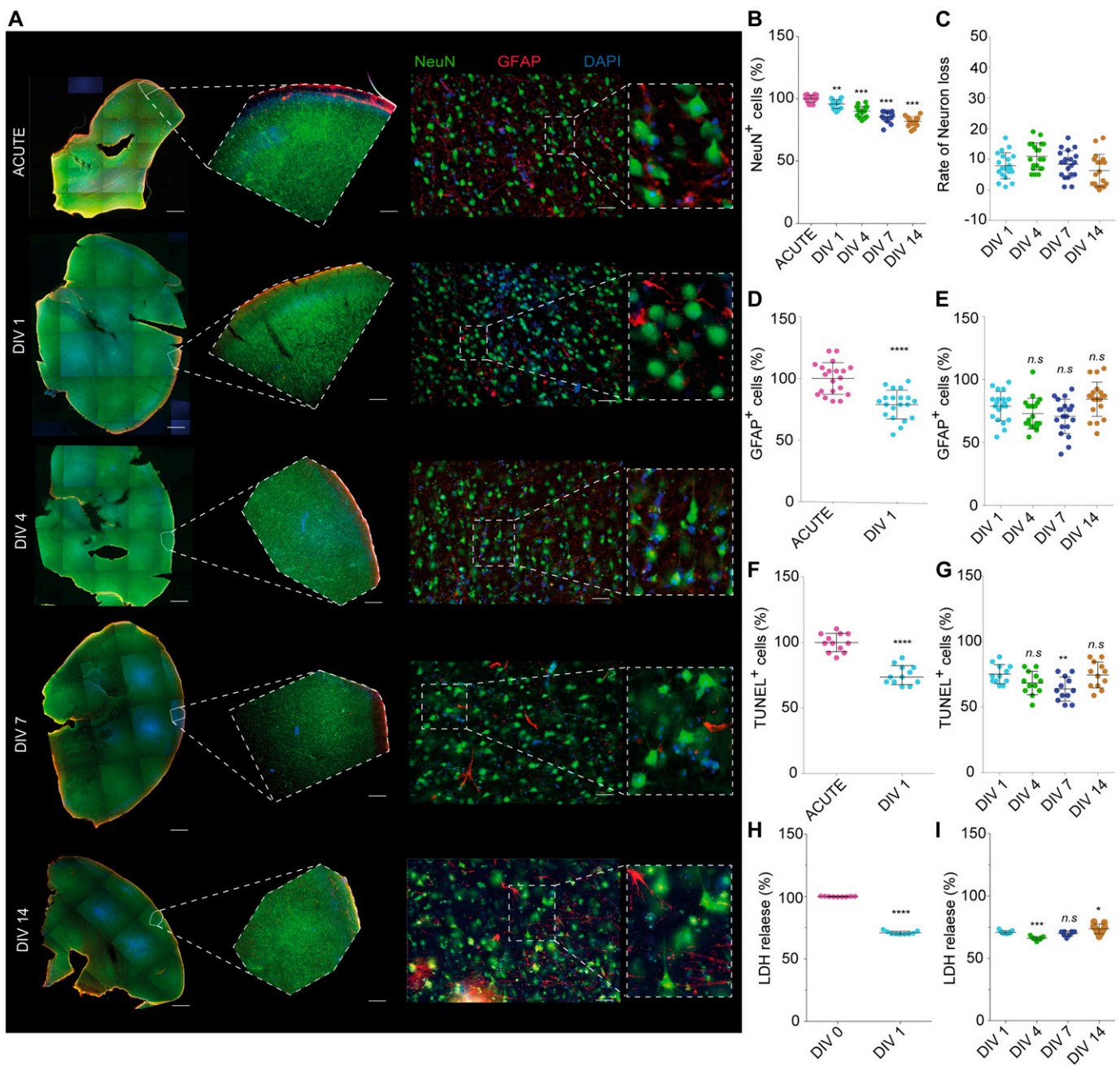

**Figure 3. Vitality quantification of human brain section model.**

Acute sections were collected from each batch as control. Sections cultivated for weeks were then compared with acute sections, with N = 20 patients. **(A)** From left to right: 5× tiled representative fluorescence images of tissue sections for Acute, DIV1, DIV4, DIV7, and DIV14. Sections were costained against the neuronal marker (NeuN), astrocyte marker (GFAP), and nuclei (DAPI Fluoromount). Scale bars are 1 mm; representative 5× image of each time point showing the cytoarchitecture of the slice. Scale bars are 200 $\mu$m; representative 20× image of each time point showing neurons (NeuN), astrocytes (GFAP), and nucleus (DAPI). Scale bars are 50 $\mu$m. **(B–E)** Quantification of NeuN (neurons) positive staining demonstrated a significant difference in the distribution of neuronal sections between fresh versus cultured sections ($P < 0.0001$). (C) Quantification of rate of neuronal loss showed a negligible reduction over DIV1–14. (D) Quantification of GFAP (astrocytes) shows an increase on fresh sections versus DIV1 because of injury ($P < 0.00001$), and (E) quantification of astrocytes loss shows a nonsignificant difference from DIV1 to 14 (DIV4 = −5.97%, DIV7 = −8.15%, and DIV14 = 5.58%, one-way ANOVA with Bonferroni correction for multiple comparisons). **(F)** Cellular necrosis in the brain slice cultures was done using a TUNEL assay for N = 3 patients at different time points. **(G)** Quantification of TUNEL-positive cells shows a nonsignificant difference from DIV1 to 14 (DIV4 = 6.74%, DIV7 = 11.35%, and DIV14 = 0.613%). **(H)** Cellular metabolism was quantified using an LDH assay in different time points. **(I)** Quantification of LDH assay shows a nonsignificant difference from DIV1 to 14 (DIV4 = 33.87%, DIV7 = 34.51%, and DIV14 = 26.31%). Statistics were performed using unpaired $t$ test with Welch's correction. *$P < 0.05$, **$P < 0.001$, ***$P < 0.0001$, ****$P < 0.00001$, ns = nonsignificant. A representative result of three independent experiments is shown (error bar represents ± SD).

−4.79%, −0.96%, and +2.77%, respectively (one-way ANOVA with Bonferroni correction for multiple comparisons, Fig 3I).

To assess whether the electrophysiological activity of the sections was maintained during the culture period, sections were cultured for 1–14 d before electrical recordings were performed (N = 6 patients). To further assess the viability of the sections, extracellular electrical activity was recorded using a Multi-Electrode array (MEA) (Fig 4A). To evoke spontaneous electrical activity, perfusion medium containing elevated levels of potassium was used. Firing activity was detected for at least 50% of electrode sites in all the recordings. Fig 4B shows the average recorded neuronal action potential for both DIV1 and DIV14. The extracellular spikes recorded from the sections revealed no changes in amplitude or firing profile as a result of being in culture (Fig 4C). In general, sections cultured for 14 d were found to generate electrical activity similar to sections that were cultured for 1 d.

To assess if the ex vivo maintenance of neuronal tissue could potentially lead to alterations in the expression profiles of cells, we made use of RNA-sequencing based gene expression analysis using 500 mg tissue (~n = 4 sections each, Fig 5A, temporal cortex). Fig 5B shows that there is no significant differential expression between the top 500 up- and down-regulated genes in the cultured versus acute sections. We identified a loss in cell-specific expression of neuronal genes between freshly cut sections and the sections after 7 d of culture, but no significant differences were seen between 7 and 14 d of culture (Fig 5C). Fig 5D–G shows the cell-class signature of the individual cell type compared between the cultured sections. The cell-specific signatures were chosen based on their fidelity score, as reported previously (Kelley et al, 2018).

In summary, we identified a significant loss of the cell-specific expression of neuronal genes between freshly cut sections and

cultured sections because of the tissue trauma, but no significant differences were seen between 7 and 14 d of culture, suggesting that sections recover and maintained stable expression profiles after the initial trauma. This was carried out by extracting cell-specific signatures of the primary cells in the CNS, including neurons, astrocytes, microglia, and oligodendrocytes, in agreement with a recently published covariation analysis (Kelley et al, 2018). The cell-class signature of microglia was found to be increased after 7 d of culture compared with the fresh sections, which then decreased to initial levels again after 14 d. Astrocyte-class and oligodendrocyte-class signatures did not differ between the fresh and cultured sections (Fig S1A and B).

## Human ex vivo GBM model

Once the sections were characterized and validated, we set out to use the model presented here to assess GBM progression in its natural environment. Based on the results of the cell death detection assays, tumor injection was performed after a resting/stabilization period of 1 d. For the tumor injection, a microsyringe mounted to a microsyringe pump was positioned and ~20,000 tumor cells (in 1 μl) from either a proneural gliobastoma stem cell line (GSC_CL1) or a mesenchymal (GSC_CL2) cell line were injected just below the upper surface of the sections (Fig 6A). Both cell lines are patient-derived primary GBM cell lines that have been extensively studied and reported upon (Heynckes et al, 2019). GSC_CL1 is an MGMT-methylated cell line, making it sensitive to temozolomide (TMZ) treatment, whereas GSC_CL2 is a non-methylated at the MGMT locus, making it resistant to TMZ treatment. Both cell lines were fluorescently tagged for ease of identification postinjection and for quantification of tumor volume. Successful tumor cell injection was validated by imaging the sections directly after injection and by further imaging at time points after 4, 7, and 14 d in culture (days postinjection [DPI]). Both cell lines show cloud-like tumor growth in the initial 2 d (data not shown), whereas by DPI3, a defined border is observed (Fig 6B, i and 6C, i). We measured the total tumor area from the time of injection to 14 d in culture and found significant proliferation of both cell lines within the cultured sections. The mesenchymal GSC_CL2 cell line shows a distinctive, highly invasive proliferation pattern (+59.2% fold change on DPI14, <0.0001) compared with the proneural GSC_CL1 cell line, which shows a much more gradual proliferation profile (+32.3% fold change on DPI14, P < 0.0001) (Fig 6D). By DPI14, we found that the mesenchymal cell type migrates beyond the boundary of the slice and infects the whole Millipore insert, whereas the proneural cell type migrates until the border of the slice and does not continue beyond the borders of the neural sections (Fig S2A and B).

## Effect of TMZ on tumor proliferation

We further tested the capability of our model to reflect changes in tumor viability under drug treatment. TMZ (Temodal/Temcad) is a chemotherapeutic drug that is widely used and established in the treatment of GBM. In contrast to many other chemotherapy treatments, TMZ can reach the brain via systemic application and is a standard in GBM therapy. At DPI4, we started a treatment regimen

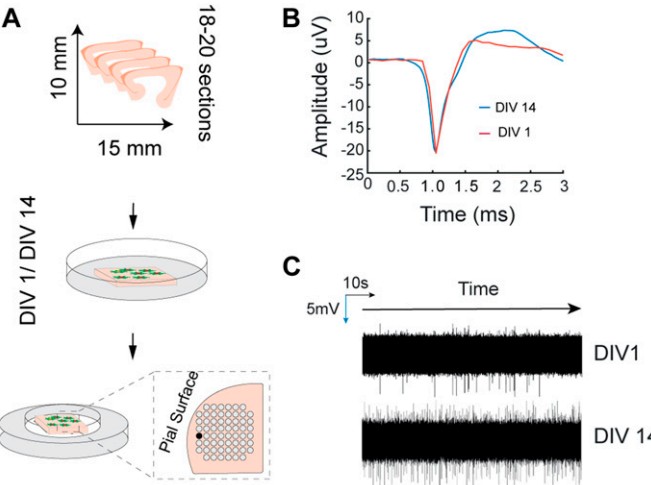

**Figure 4. Electrophysiological activity profiling by means of multi-electrode array.**
**(A)** Illustration of the experimental workflow. Sections were cultured for up to DIV14, and recordings were performed to assess electrophysiological activity. The sections were placed on the recording array as illustrated. Neuronal activity was evoked by means of perfusion with high K⁺ medium. **(B)** Recorded data were high-pass filtered (300 Hz), and neuronal events were extracted and averaged before plotting (red: DIV1, blue: DIV14). **(C)** Random sampling of the extracellular electrophysiological recording.

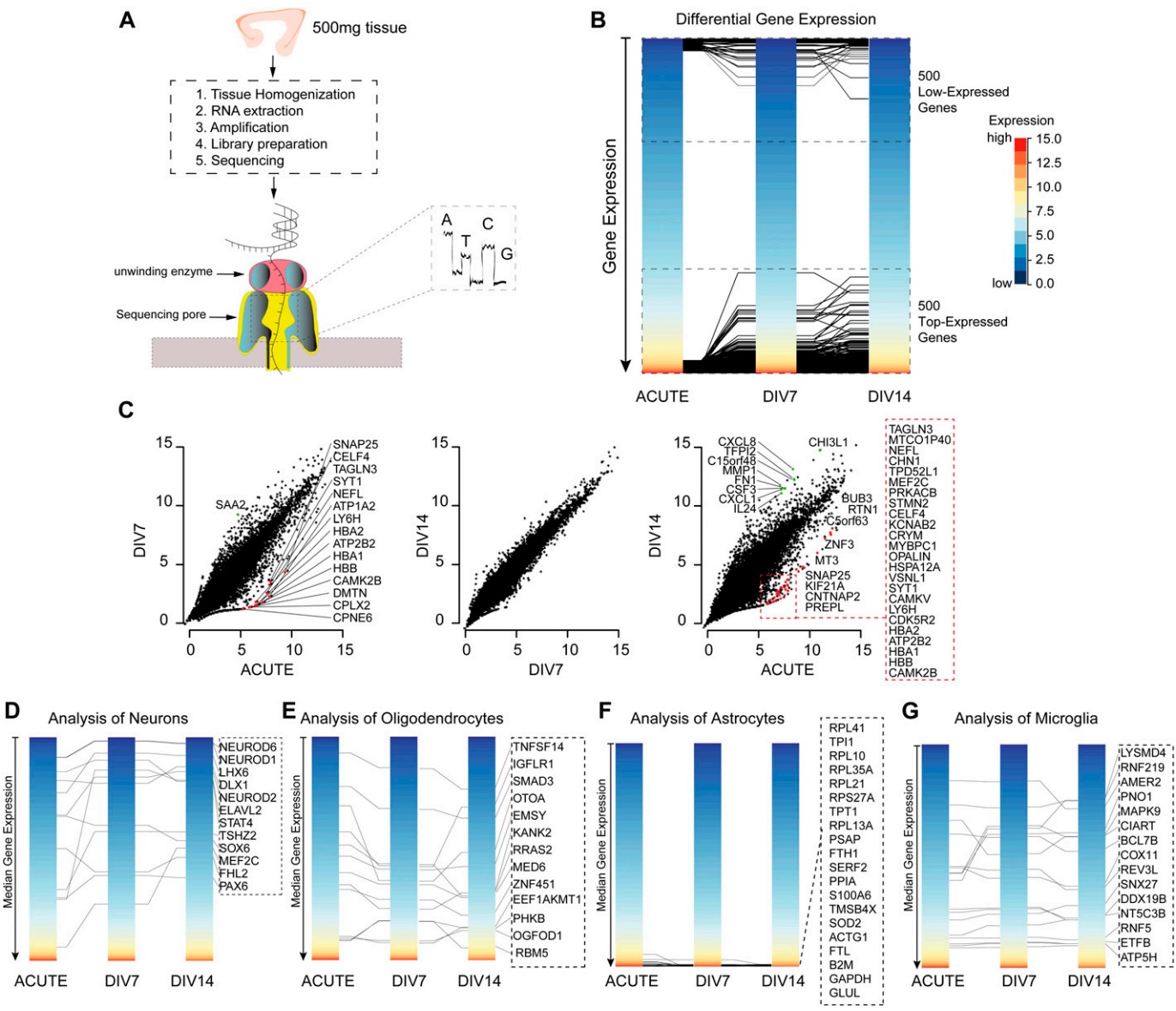

**Figure 5.   Gene expression analysis.**
**(A)** Workflow of an RNA-seq experiment with ~500 mg tissue using Nanopore MinION. After the culture period specified, the tissue was homogenized, RNA extracted, amplified, the library prepared, and sequenced from N = 3 patients. **(B)** Differential bulk gene expression showing top 500 up- and down-regulated genes. The top 500 up- and down-regulated genes lay within the same expression region. **(C)** Comparison of the gene expression profile at different time points. The gene expression profile exhibits a change in expression from the acute to DIV7. However, the expression profile remains stable when compared between DIV7 and DIV14. **(D–G)** Cell-specific expression profiles of neurons, oligodendrocytes, astrocytes, and microglia. Key genes from each cell type are shown in the dotted box next to the expression plot. The cell-specific signatures were chosen based on their fidelity score. The scale bar for y-axis is global as represented in (B).

with 50 $\mu$M TMZ, as previously reported (Borges et al, 2011; Ostermann et al, 2004). To ensure consistent concentration of TMZ, the medium was replaced every second day. Tumor size measurements and quantification revealed reduced/halted tumor growth under TMZ treatment for the TMZ-sensitive cell line (GSC_CL1), whereas the TMZ-resistant cell line (GSC_CL2) showed a minor response to TMZ treatment (Fig 6B, ii and 6C, ii). These results agree with previous reports on the mechanism of action of TMZ on the proliferation of the GBM (Schaub et al, 2018). Taken together, our data propose that the slice model technique can be used to

investigate the impact of chemotherapeutics able to cross the blood–brain barrier on tumor growth.

## Glioma cell migration and penetration

It has been reported that GBMs primarily follow the white matter where infiltrative invasion of the brain was observed (Engwer et al, 2015; Esmaeili et al, 2018). We validated this hypothesis by injecting GBM cells into the cortical portion of our sections. After 7 d, we observed that GSC_CL1 showed an invasion only into white matter

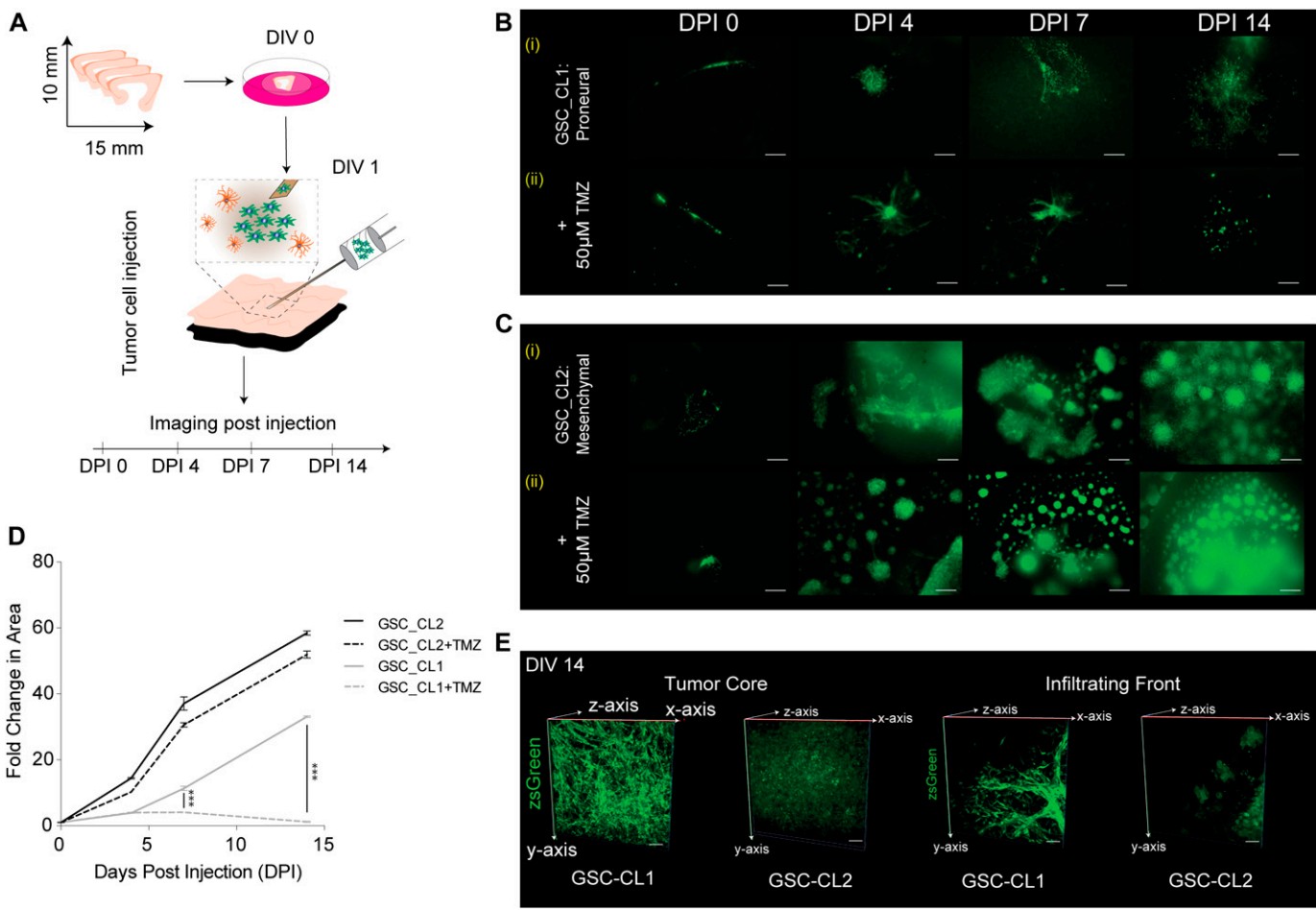

**Figure 6. Human GBM model.**
**(A)** Illustration of the experimental workflow showing tumor injection on DIV1, and tumor progression was further validated using imaging. **(B)** (i) proliferation pattern of the TMZ-sensitive proneural GSC_CL1 over 14 d of culture. (ii) The proliferation pattern is disrupted when TMZ (50 $\mu$M) is added to the culture medium. TMZ was added to the medium 3 d postinjection of GBM cells into the slice (DIV4). **(C)** Scale bars are 200 $\mu$m. (C) (i) proliferation pattern of the mesenchymal cell type GSC_CL2 over 14 d. (ii) The proliferation is uninterrupted when TMZ (50 $\mu$M) is added to the culture medium. TMZ was added to the medium 3 d postinjection of GBM cells into the slice (DIV4). Scale bars are 200 $\mu$m. **(D)** Quantification of the increase in the area of the injected tumor for GSC:CL1 and GSC:CL2 shows +59.2% fold change for GSC:CL2 while +32.3% for GSC:CL1, $P < 0.0001$, and in the presence of 50 $\mu$M TMZ, GSC_CL1 shows halted growth, whereas the GSC_CL2 shows a minor response to the treatment. **(E)** 3D imaging and reconstruction was performed using two-photon microscopy. The left images demonstrate the tumor core in GSC_CL1 and GSC_CL2, and the right images represent the infiltrative margin of the tumor for GSC_CL1 and GSC-CL2. Scale bars are 50 $\mu$m. Unpaired $t$ test with Welch's correction were used.

regions and avoided the cortex. In GSC_CL2 cells, we observed a growth pattern based on clonal expansion within both the white matter and the cortex (Fig S3A). Two-photon imaging was made use of to quantify the invasive front and penetration depth of the injected GBM cells. Quantification and reconstruction of the 3D volume of the tumor mass from both the "tumor core" and the "invasion front" was carried out (Fig 6E, Two Photon video in Videos 1 and 2). In addition, the cultured sections exhibited strong vascular networks when stained for collagen IV antibody even after 14 d ex vivo. Our finding of glioma cells (GSC_CL1 and GSC_CL2) migrating along the blood vessel is similar to previously reported work (Farin et al, 2006) (Fig S4A, Videos 3, 4, 5, and 6). Quantification of the collagen[+] vessels shows that the GSC_CL1 has a greater number of intersections (Fig S4B, iii).

To further validate these claims, the culture medium was collected from both control and GBM-injected sections to quantify

changes in cytokine levels because of the tumor microenvironment. We report a significant increase in anti-inflammatory cytokines (G-CSF) and pro-inflammatory signal molecules (TGF-$\beta$) (Fig 7) in both cell lines compared with the control conditions.

## Astrocyte purification from cultured sections

The TME consists of a variety of cells playing a crucial role in tumor growth, metabolism, and the immune environment. Increasing evidence points to the crucial role of the cellular environment in tumor therapy; therefore, it is important to investigate these cells in their native state. The model we present allows us to explore the interactions of a variety of cell types without resorting to a 2D mono- or co-culture model (as shown in Fig S3B). To investigate the transcriptional profile of astrocytes in the TME, we specifically isolated astrocytes using magnetic-assisted cell sorting (MACS)

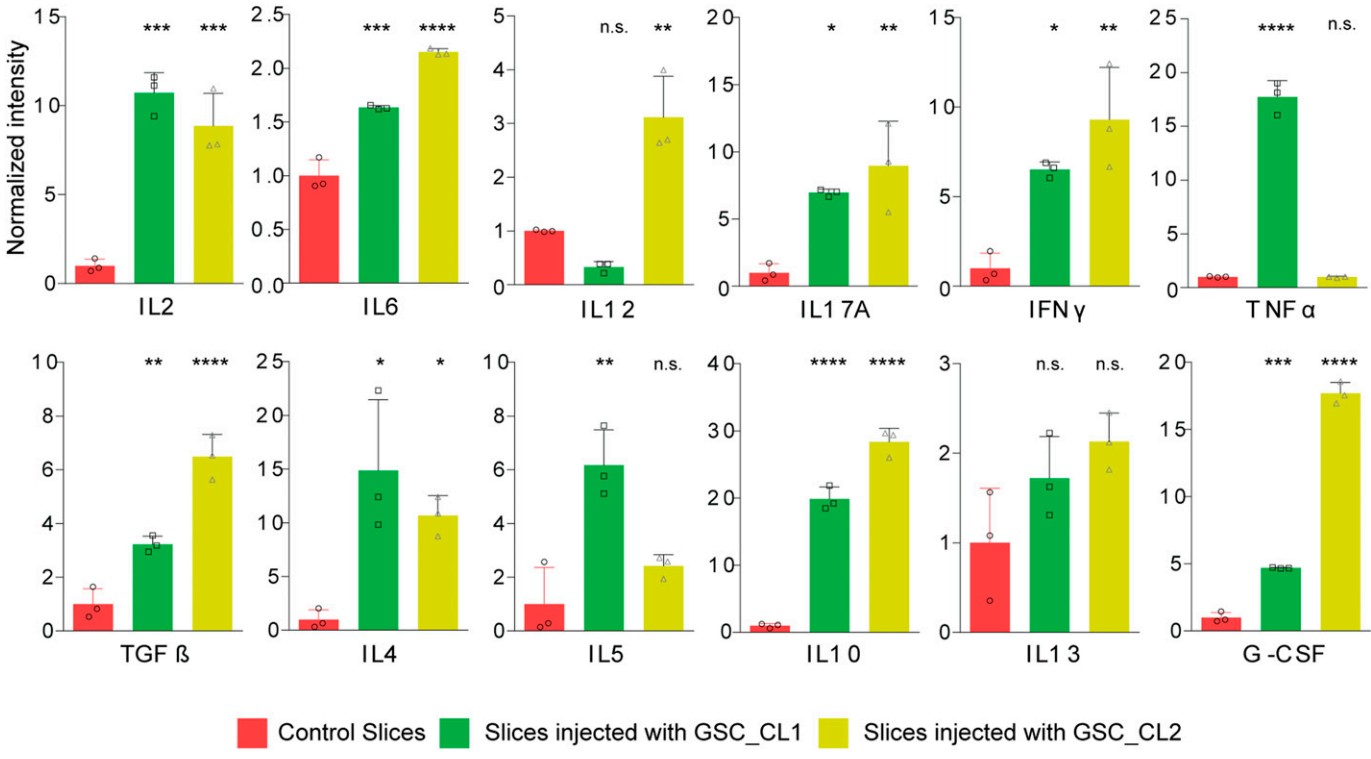

**Figure 7. Cytokine profile of different GBM cell types: cytokine environment of sections with and without tumor injection.**
The cytokine profiling shows that there are significant changes in sections because of the development of the TME using one-way ANOVA test with Bonferroni's multiple comparison test. *$P < 0.05$, **$P < 0.001$, ***$P < 0.0001$, ****$P < 0.00001$.

from both control and tumor-injected sections (GSC_CL1) (Fig 8A) and performed comparative analyses with astrocytes derived from patient tumor core and healthy cortex (Fig 8B). The purity of astrocytes extracted using GLAST antibody was verified using FACS to show that extracted cells are indeed astrocytes and showed extremely low contamination of tumor cells (pZsgreen) (Fig S5).

To screen the genes that were differentially expressed because of tumor infiltration, we pooled publicly available datasets from both human and murine astrocytes from a recently published classification, which distinguishes between an inflammatory (termed "A1") astrocyte subtype and a noninflammatory subtype (termed A(n) [Liddelow et al, 2017]), which was also reported to be present in stroke-affected cortical specimens (Zamanian et al, 2012). To supplement these findings, reported signature genes were chosen and compared with human astrocyte datasets, resulting in a reduced gene set of 17 genes, which revealed stable expression in human astrocytes compared with their murine counterparts (measured by their fidelity score). We then analyzed the expression of these genes, separated into a Pan-activation, A1-specific, and A(n)-specific gene set of our purified astrocytes from both fresh tissue specimens and our slice culture model sections (Fig 8C). The astrocytes revealed a stable expression of genes that belong to the noninflammatory subtype that corresponds with RNA-sequencing expression data from the published datasets (Zhang et al, 2016; Kelley et al, 2018).

In summary, our results indicate that our tumor injection model in neocortical sections can simulate reactive changes in astrocytes similar to those found in primary GBM tissue. With the presented model, it becomes possible to investigate individual components of the microenvironment and to obtain new insights into cell-specific functions in the maintenance of GBM malignancy.

## Discussion

In the last few decades, one of the major challenges in translational neuro-oncology research has been the development of GBM environment models that can paint an accurate picture of the microenvironment of the CNS, that are highly reproducible, easy to use, and widely available. In recent years, a plethora of models have been developed that have individually contributed to enormous growth in our knowledge about malignant brain tumors. The commonly reported in vitro monolayer cell cultures of GBM are limited in their ability to recapitulate every aspect of the TME (Wu & Swartz, 2014). This was followed by the use of 3D collagen hydrogel models that fail to replicate the true ECM composition of neuronal tissue and mimic the normal brain tissue/tumor environment (Grotzer et al, 2016). Furthermore, research into the mechanism of cancer pathology and cancer drug development have relied heavily on genetically modified, cell line xenograft, and, more recently, patient-derived xenograft mouse models (Jackson & Thomas, 2017). These mouse models are globally accepted gold standards but must be used with caution because they exhibit not just altered

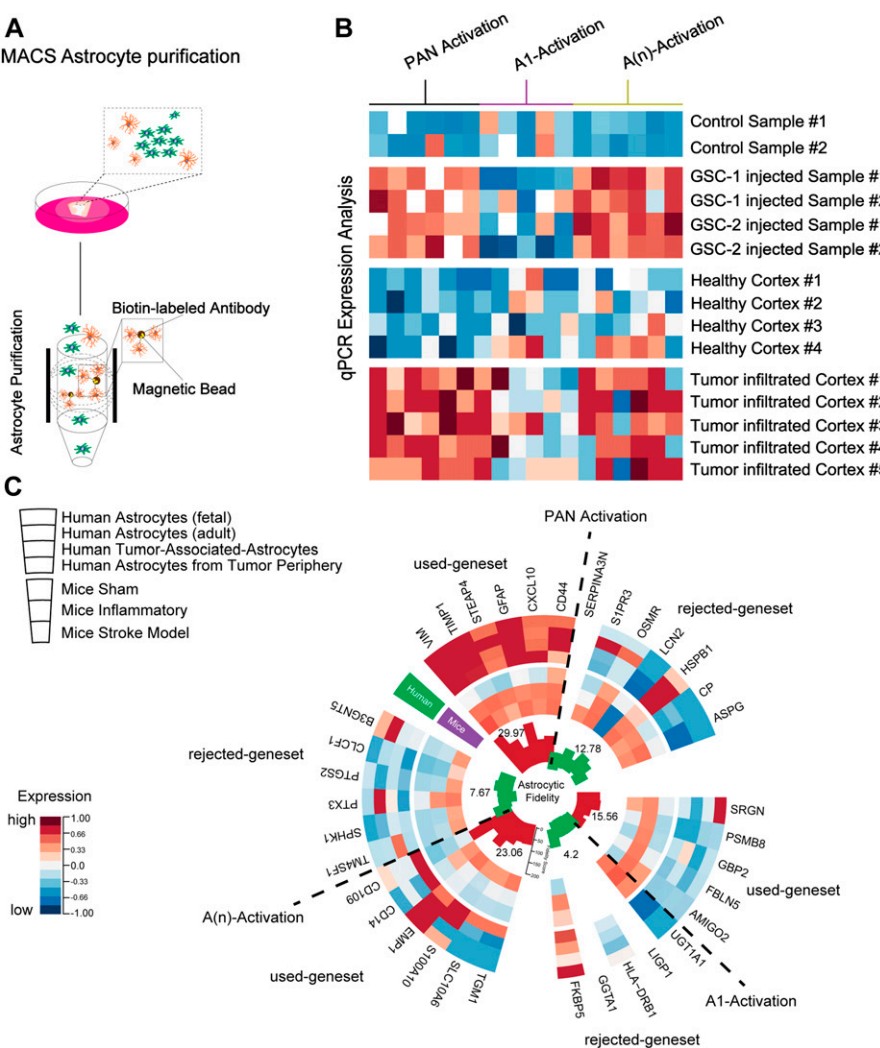

**A**
MACS Astrocyte purification

**B**

**C**

**Figure 8. Astrocyte profiling using ex vivo model.**
**(A)** Astrocyte extraction protocol from the cultured sections. MACS was used to extract the astrocytic cells with biotin-labelled antibody. **(B)** qPCR-based heat map of signature astrocytic genes that mark the different reactive state of the extracted astrocytes. Signature genes were extracted from publicly available datasets. **(C)** Expression analysis of reactive marker genes selected by specificity for humans and astrocytic fidelity score, in astrocytes purified from GBM patients and our human slice model with tumor injection.

immune responses but also significant interspecies differences. To try and overcome these limitations, we have established this method to viably preserve healthy human cortical sections for several weeks, preserving 3D anatomical structure and cellular complexity. Living brain tissue, in particular derived from human brain, is the closest representation of the in vivo human brain outside of a living person and is an ideal matrix for studying malignant glioma invasion ex vivo. Because live human neuronal tissue is poorly characterized, before proceeding to make use of the sections to study diseases like GBM, the vitality of sections was confirmed using different approaches.

Immunohistochemical evidence shows that serum was not needed to maintain the cytoarchitecture of the healthy human cortical explants for up to 2 wk. The main key players such as astrocytes and neurons were well preserved for the entire duration of culture without a substantial change in numbers for up to 2 wk with no exogenous growth factors. To verify if there was any alteration in the expression profile, chosen signature genes were pooled together and compared between acute and DIV7 and DIV14. This revealed mathematically significant differential expression

between cultured and fresh sections (Fig S1), which did not translate into measurable effects during the culture period. These data were further supported by electrophysiological experiments, which showed that network activity in the cultured sections was maintained even after 2 wk in culture. In summary, the slice model reported here is the closest representation to in vivo conditions to date.

### Establishment of an ex vivo human model to study GBM

Until now, research groups have used the technique of seeding tumor spheroids on top of the human neural tissue sections (Jung et al, 2002), which shows reduced invasiveness, as the cells migrate on the slice surface instead of penetrating into the tissue (Eisemann et al, 2018). Here, we injected two well-characterized GBM cell lines, which showed that the invasive behavior of each cell line is profoundly different, and the invasion pattern is similar to in vivo reports, migrating toward the white matter tract with the proneural cell type (Wang et al, 2019). This contributes to the understanding of the biological dynamics of infiltrating GBM cell lines

through healthy brain sections. We maintained the anatomical structure of the sections after tumor injection. Tumor progression in the brain slice was dramatic, even after a week in culture.

To further demonstrate its utility as a preclinical model, (i) we tested the response of the growing tumor mass using a commercially available chemotherapeutic agent TEMODAL (TMZ) (Wick et al, 2009). A distinct arrest and reduction in the proliferation was seen in methylated MGMT (GSC_CL1) compared with the non-methylated MGMT (GSC_CL2) glioma cell type, as reported previously (Schaub et al, 2018). (ii) We show that human brain sections exhibit a dense vascular network positive for collagen IV (with and without tumor injection), although there is a lack of vascular supply to the tissue in the sections, similar to a previously reported rodent model (Moser et al, 2003). Thus, the method presented here provides a simplified model for assessing responses of different GSC cell lines to study the brain microenvironment.

As the investigation of the cellular environment in GBM is a major challenge in most available GBM models, the extent of knowledge about reactive changes of nonmalignant cells in the tumor environment is very limited. One goal was to examine to what extent our model exhibits changes in the nonmalignant cellular environment of GBM, and if so, whether it can be used to address this lack of knowledge. The TME is crucial for the progression of tumors (Guan et al, 2018), with 40% being astrocytes (Zhang et al, 2016) and 10–15% being microglial cells (Placone et al, 2016), and plays a vital role in progression, invasion, and angiogenesis of the tumor mass. Therefore, we aimed to test our model by analysis of reactive changes of astrocytes in the tumor environment. MACS allowed us to purify astrocytes from ex vivo sections before and after GBM injection. The degree of serum-induced astrocyte differentiation is eliminated (Zhang et al, 2016) because the sections were always grown in serum-free media. Our imaging data revealed that the sections with injected GBM cell lines maintained their vitality even after a week in culture (Fig S3B). At the transcriptomic level, our data corroborate differences between astrocytes in human and murine models in different reactive states, showing evidence that the TME cannot be sufficiently modeled using murine models. We also found that our model showed a significant increase in cytokines and neurotropic factors, such as IL-10, TGF-$\beta$, and G-CSF, which are responsible for tumor progression and glioma genesis compared with the control environment. Our technique will also allow one to purify a variety of cell types, such as oligodendrocytes, neurons, or microglia cells, which could be profiled in parallel from the same neocortical sections.

In this highly evolving era of neuro-oncology research, defining an in vivo signature to study tumor invasion and TME remains challenging. Therefore, we optimized an ex vivo model that can reliably and quantitatively measure GBM proliferation, in an ECM similar to what is found in the human brain. The advantage of this model is that it uses otherwise discarded tissue to create an experimental model that closely represents what actually occurs in humans in vivo rather than using rodent tissue to avoid cross-species reactivity (Huszthy et al, 2012). It is relatively simple and requires minimal media components, without any extraneous media or growth factors.

However, this model also has some limitations. Because access to the human tissue specimens is limited to therapeutically motivated cases, it is therefore necessary to establish proper surgical protocols to ensure collection and processing of the resected tissue with minimal damage to maintain the structural integrity of the tissue sections. We also need to have a large sample size of patient donors to be able to experimentally verify with confidence the effects that are seen because of treatment because the resection region is dependent on the location of the tumor. Also, when primary patient-derived cell lines are used for GBM modeling, we can have inflammatory activation of immune cells as a result of infiltration by foreign cells. Finally, studies that involve the blood–brain barrier cannot be carried out using our present model.

In conclusion, the present study demonstrates the feasibility and effectiveness of an interface method to maintain human explants for prolonged periods. Our study extends the functional applications of the model by investigating the proliferation of GSC cell lines in an ECM similar to that which is found in the human brain and studying the role of purified astrocytes in the TME. This model therefore has potential applications to the fields of neuroscience, neuro-oncology, and pharmacotherapy.

# Materials and Methods

Permission to use human brain tissue for evaluation, imaging procedures, and experimental design was obtained from the local ethics committee of the University of Freiburg (Approval 100020/09 and 5565/15). Human brain tissue specimens were obtained with informed consent under a Declaration of Helsinki, as requested by the local ethics committee (No. 187/04). Human neocortical tissues (N = 26) were sampled from planned regions, either tumor core or distant cortex without tumor filtration (n = 21) (at least 2 cm away from the tumor core) or cortex from epilepsy surgery guided by intraoperative neuronavigation (n = 5) (Cranial Map Neuro Navigation Cart 2; Stryker) during resection. The healthy cortical tissue was assessed by EEG and MRI. Sparing use of cauterization during surgery provided a higher quality of organotypic sections, judged by culture survival and ease of recording from neurons. Detailed information regarding the donors is provided in Table S1. Tissue was immersed in 4°C Neurobasal Medium (Lot No. 1984948; Gibco) immediately postresection and transferred to the laboratory for tissue dissociation/sectioning.

### Presectioning preparation

A sterile working field was set up for the sectioning procedure. This is to avoid any contamination of the tissue being processed for the sectioning. All tools required for this procedure were sterilized following clinical protocols and placed within reach of the experimenter to prevent delays. The sections were prepared using a vibratome (VT1200; Leica Biosystems). The sectioning chamber was filled with ice-cold preparation medium containing Hibernate-A Medium (Lot No. 1994548; Gibco) supplemented with 13 mM D-glucose (Lot No. SLBX3638; Sigma-Aldrich), 30 mM N-methyl-D-GLucamin (M2004; Sigma-Aldrich), and 1 mM GlutaMAX (Lot No. 1978435; Gibco), saturated with carbogen (95% $O_2$ and 5% $CO_2$) for 10

min before the sectioning of resected tissue. Millicell inserts (No. PICM03050; Millipore) were placed in each well of a six-well culture dish with 1 ml of growth medium containing Neurobasal L-Glutamine (Lot No. 1984948; Gibco) supplemented with 2% serum-free B-27 (Lot No. 175040001; Gibco), 2% Anti-Anti (Lot No. 15240-062; Gibco), 13 mM D-glucose (Lot No. RNBG7039; Sigma-Aldrich), 1 mM $MgSO_4$ (M3409; Sigma-Aldrich), 15 mM Hepes (H0887; Sigma-Aldrich), and 2 mM GlutaMAX (Lot No. 1978435; Gibco).

## Preparation of human brain section cultures

Capillaries and visibly damaged tissue were dissected away from the tissue block while being submerged in preparation medium. 300-$\mu$m-thick sections were obtained at 0.12 mm/s and were incubated in cold preparation medium (4°C) for 10 min before plating. Tissue blocks (1 cm × 2 cm) typically permits preparation of 18–20 sections. One to four sections were gathered per insert, with care to prevent them from touching each other, depending on the experiment. The obtained sections were cultured in a humidified incubator at 5% $CO_2$ and 37°C for a maximum of 14 d postresection. The collected medium was frozen at −20°C for ELISA-based measurements. From each donor sample, three to four sections were freshly fixed postsectioning in 4% PFA and used as control specimens.

## ELISA measurements of cytokine and LDH release assays

Culture medium was collected from each insert containing four sections, and the cytokine profiling assay and the LDH assay were carried out according to the manufacturer's instructions.

For the LDH assay, 50 $\mu$l of the collected medium was mixed with 50 $\mu$l of CytoTox 96 Substrate Reagent (CytoTox 96 Non-Radioactive Cytotoxicity Assay kit) in each well. After 30 min of incubation in the dark, 50 $\mu$l of stop solution was added to each well. Finally, the absorbance was read at 490 nm using a plate reader (Tecan). LDH release percentage of control was calculated as $(OD_{sample}/OD_{control}) \times 100$. The analysis was performed from three wells (each containing four sections) per time point for the LDH assay. Data are reported as mean value ± SD for continuous variables and were further analyzed by one-way ANOVA. The cytokines from the supernatant (IL-2, IL-4, IL-5, IL-6, IL-10, IL-12, IL-13, IL-17A, IFN$\gamma$, TNF-$\alpha$, G-CSF, and TGF-$\beta$1) were determined using Multi-Analyte ELISArray kit (QIAGEN GmbH) in accordance to the manufacturer's instructions for the control and tumor-injected samples.

## TUNEL assay

The TUNEL assay (Roche) was conducted according to the manufacturer's protocol and combined with immunohistochemical labelling. Briefly, 550 $\mu$l of buffer (reagent A) was mixed with 50 $\mu$l of enzyme solution (reagent B). Each slice was incubated with 100 $\mu$l of this solution at 37°C for 1 h, protected by a coverslip. Immunohistochemical staining was conducted afterward as described above.

## Viral transduction by constitutive reporter lentiviral vectors

For whole-cell tracking, primary cultured GBM cells were transduced with lentiviral particles (pZsGreen1-1 Vector, Takarabio; Clonetech). For the transduction, 5 × 10^5 cells were seeded per well and incubated overnight in an incubator at 37°C and 5% $CO_2$. Quantification of the particles was calculated according to the manufacturer's instructions. The transduction mix was prepared by adding the required volume of thawed viral particles and Polybrene (800 $\mu$g/ml). Medium was changed after 1 d. Quality of transduction was measured after 2 d by microscopic observation.

## Tumor invasion model

By using pZsGreen transfected GBM cells, it was possible to evaluate the invasion profile in the brain sections (Fig 3B). Post-trypsinization, a centrifugation step (5,000$g$, 10 min), was performed, followed by harvesting the cells and suspending them in MEM at 20,000 cells/$\mu$l. The cells were then used immediately for injection into tissue sections 24 h postplating (37°C, 5% carbogen). The cells were then loaded into a 10-$\mu$l syringe, driven by a microliter syringe pump for accurate dispensing of tumor cells. The injection tip was positioned just below the top surface of the cultured sections, and ~20,000 cells were injected (in 1 $\mu$l). A 10-$\mu$l Hamilton syringe was used to inject 1 $\mu$l of cells into the white matter portion of the cultured sections. Sections with injected cells were incubated at 37°C and 5% $CO_2$, and fresh culture medium was added every 48 h.

Migration of tumor cells into the surrounding normal brain tissue was observed on DIV1, DIV4, and DIV7 using fluorescence and two-photon imaging. The brain sections were imaged using an upright two-photon microscope Olympus FV1000 with Mai Tai Deep See Laser (Spectra Physics; Newport Corporation) under a 20× lens objective (XLUMPLFLN, NA = 1). The two-photon imaging were carried out using an excitation wavelength of 870–890 nm, with an emission filter of 515–560 nm used to image the GFP signal and an emission filter of 590–650 nm to detect the Alexa-Fluor 555 signal. Images from the cultured neural sections were acquired with 1- to 2-$\mu$m z-axis increments and 800 × 800-pixel resolution.

Tumor proliferation was also monitored by regular fluorescence imaging at days 0, 4, and 7 by means of an inverted microscope (Observer D.1; Zeiss). After 7 d of incubation, sections were fixed and used for immunostaining or astrocyte extraction.

## Immunohistochemistry and quantification

The sections with and without tumor injection were first fixed in 4% PFA and then permeabilized using 0.5% Triton (TX-100) overnight at 4°C. Blocking was performed using 20% BSA, supplemented with 1% Triton for 4 h. The sections were then incubated in primary antibodies: anti-NeuN (mouse, 1:1,000, MAB377; EMD Millipore), anti-GFAP (rabbit, 1:2,000, G9269; Sigma-Aldrich), and anti-collagen IV (ab6586; Abcam) and incubated overnight at 4°C. The sections were then labelled with secondary antibodies Alexa 488 and Alexa 555 for 3 h at RT. The sections were then mounted on glass slides using DAPI Fluoromount (Cat. No. 0100-20; Southern Biotech). Quantification was carried out based on the cell type, in ImageJ (Schindelin

et al, 2012). Four to six representative fields were acquired per section image (20× magnification). After background subtraction and auto-thresholding using the ISODATA algorithm, total number of cells were counted using ImageJ -> "Analyze Particles." One-way ANOVA followed by Bonferroni correction for multiple comparisons test was performed for all experimental groups using GraphPad Prism (GraphPad Software Inc.). To calculate the area covered by tumor in injected sections, regions of proliferation were detected by thresholding the images. The area of this region of interest was then measured using ImageJ and further analyzed.

Blood vessels were traced using Simple Neurite Tracer, and intersections were detected using Sholl analysis v4.0.0 with enclosing radius cutoff as 1 and Sholl methods as linear as described previously (Theer et al, 2014).

## Flow cytometry

Tissue specimens were mechanically dissociated using a glass homogenizer on ice and sequentially passed through 100- and 40-$\mu$m nylon cell strainers (No. 352360 and No. 352340; BD Falcon). The mesh was then rinsed several times with 4°C cold PBS/EDTA. The resulting cell suspensions can be kept on ice for up to 20 min while other tissue samples are being processed. After centrifugation (310$g$; 4°C; 6 min) and removal of the supernatant, the cell pellet was resuspended in 5 ml of −20°C cold 80% methanol, added drop-wise under constant gentle vortexing. Samples were incubated for 30 min on ice and subsequently overnight at −20°C before being subjected to staining. Alternatively, the samples can be stored at −20°C for up to 1 yr.

The cell suspension was washed and centrifuged at 350$g$ for 5 min and $2 \times 10^6$ cells were used. The cells were resuspended using permeabilization buffer (0.1% Triton X-100 in 1× PBS) for 5 min at room temperature. Samples were centrifuged briefly, and the pellets were washed two times with 1× PBS. 5 $\mu$l of TruStain FcX was added per million cells in 100 $\mu$l staining volume to avoid unspecific antibody binding. It is not necessary to wash the cells between these blocking and immunostaining steps. The cells were stained with fluorochrome-conjugated antibodies such as anti-GLAST-APC along with DAPI. Antibody staining was performed according to the manufacturer's instructions. Finally, cells were washed and resuspended in at least 0.5 to 1 ml of FACS buffer depending on the number of cells. We used a Sony SP6800 Spectral Analyzer and recorded 100,000 events per sample in standardization mode with PMT voltage set to maximum to reach the saturation rate below 0.1%.

## Astrocyte separation using MACS

Sections were processed into a single-cell suspension by means of mechanical dissociation using a glass homogenizer, in HBSS with 10 mM glucose and 10 mM Hepes. To obtain a yield of 20 $\mu$g/$\mu$l of RNA, three to four sections were pooled. The single-cell suspension was cleaned using 37.0% Percoll and then centrifuged. We then made use of a cell isolation technology, based on nanoscale immuno-magnetic beads, combined with selective columns (MACS; Miltenyi Biotec). We performed a positive selection and labelled astrocytes with 20 $\mu$l of biotin-labelled anti-GLAST antibody, with $10^7$ cells used

as the input. After incubation with the first antibody, we performed magnetic labelling with 20 $\mu$m beads. The astrocytes were then purified in a MACS column, washed eight times, and the resulting output was collected into Qiazol followed directly by RNA extraction and validation using a NanoDrop.

## Electrophysiological recordings

Extracellular recordings were made using an MEA 1060 UP (Multi-Channel Systems). The sections were cut out of the inserts including the membrane and flipped face down onto the recording electrode. A slice grid (HD5; Ala Scientific) was further placed over the slice to ensure contact with the electrode array. The sections were initially perfused with physiological artificial cerebro spinal fluid (aCSF) (concentration in mM: 11 mM glucose, 25 mM NaHCO$_3$, 126 mM NaCl, 3.5 mM KCl, 1.2 mM NaH$_2$PO$_4$, 1.3 mM MgCl$_2$, and 2 mM CaCl$_2$) at 3 ml/min during the stabilization and baseline phase (60 min) followed by a high K$^+$ aCSF (11 mM glucose, 25 mM NaHCO$_3$, 126 mM NaCl, 7 mM KCl, 1.2 mM NaH$_2$PO$_4$, and 2 mM CaCl$_2$) perfused into the system for 60 min. The high K$^+$ aCSF was used to evoke activity from the tissue, which was then further analyzed offline using a custom MATLAB (Mathworks Inc.) script, as reported previously (Joseph et al, 2018). Briefly, after high-pass filtering of the data (300 Hz), four parameters were used to classify a threshold crossing event as an action potential: amplitude, falling and rising slopes of depolarization/repolarization, the hyperpolarization amplitude, and the time delay between the depolarization peak and hyperpolarization peak.

## RNA sequencing

The purification of mRNA from total RNA samples was achieved using the Dynabeads mRNA Purification kit (Thermo Fisher Scientific). The subsequent reverse transcription reaction was performed using SuperScript IV reverse transcriptase (Thermo Fisher Scientific). Preparation for RNA sequencing was carried out using the low-input PCR barcoding kit and the cDNA-PCR Sequencing kit (Oxford Nanopore Technologies), which were used as recommended by the manufacturer. RNA sequencing was performed using the MinION sequencing device, the SpotON Flow Cell, and MinKNOW software (Oxford Nanopore Technologies). Only D$^2$-Reads with a quality score above eight were used for further alignment. The quality score is measured based on the base-calling algorithm albacore (Nanopore), defined as −10 × log10 (probability of incorrect base call). Reads were rearranged in accordance to their barcode and trimmed using Porechop (https://github.com/rrwick/Porechop), with alignment performed using minimap2 (https://github.com/lh3/minimap2) and post-processed by sam-tools. Mapped reads were normalized by DESeq (Love et al, 2014). The expression matrix was analyzed with AutoPIPE (https://github.com/heilandd/AutoPipe), a supervised machine-learning algorithm and visualized in a heat map as described previously (Henrik Heiland et al, 2019).

## Statistical analysis

We routinely show the SD to allow direct evaluation of variability and differences between values in plots. Sample size is always

indicated in the figure legend and, when appropriate, *P* values are shown. All experiments were performed in triplicate, and analysis and plotting were carried out using GraphPad prism (GraphPad Software Inc.) or R statistical environment (https://www.rstudio.com/).

## Data Availability

RNA-sequencing data are available at accession code: GSE132954. Further information and request for resources, raw data, and reagents should be directed and will be fulfilled by VM Ravi, vidhya.ravi@xuniklinik-freiburg.de, and DH Heiland, dieter.henrik.heiland@uniklinik-freiburg.de.

## Supplementary Information

## Acknowledgements

The authors thank Jonathan Göldner for his continuous support in the laboratory. DH Heiland is funded by the German Cancer Society (Seeding Grand TII), Müller-Fahnenberg Stiftung, and Familie Mehdorn Stiftung. VM Ravi and K Joseph were partially funded by the BMBF-projects NEUROPHOS (13GW0155C) and FMT (13GW0230A).

### Author Contributions

VM Ravi: conceptualization, data curation, formal analysis, supervision, validation, investigation, visualization, methodology, project administration, and writing—original draft, review, and editing.
K Joseph: data curation, formal analysis, validation, investigation, visualization, methodology, project administration, and writing—original draft, review, and editing.
J Wurm: methodology.
S Behringer: methodology.
N Garrelfs: methodology.
Pd Errico: methodology.
Y Naseri: methodology.
P Franco: methodology.
M Meyer-Luehmann: resources, methodology, and writing—review and editing.
R Sankowski: methodology.
MJ Shah: methodology.
I Mader: writing—review and editing.
D Delev: writing—review and editing.
M Follo: resources, validation, and writing—review and editing.
J Beck: validation, methodology, and writing—review and editing.
O Schnell: validation, methodology, and writing—review and editing.
UG Hofmann: resources, supervision, funding acquisition, validation, and writing—review and editing.
HD Heiland: resources, data curation, formal analysis, visualization, methodology, and writing—review and editing.

### Conflict of Interest Statement

The authors declare that they have no conflict of interest.

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
