## [Reviewer comments · Life Science Alliance]

Life Science Alliance

Human Organotypic Brain Slice Culture: A novel framework for environmental research in Neuro-oncology

Vidhya M. Ravi, Kevin Joseph, Julian Wurm, Simon Behringer, Nicklas Garrelfs, Paolo d'Errico, Yashar Naseri, Pamela Franco, Melanie Meyer-Luehmann, Roman Sankowski, Mukesch Johannes Shah, Irina Mader, Daniel Delev, Marie Follo, Jürgen Beck, Oliver Schnell, Ulrich G. Hofmann, Dieter Henrik Heiland

DOI: <https://doi.org/10.26508/lsa.201900305>

Corresponding author(s): Vidhya Ravi, University Hospital Freiburg

Review Timeline:

Submission Date:	2019-01-14
Editorial Decision:	2019-02-12
Revision Received:	2019-05-13
Editorial Decision:	2019-06-06
Revision Received:	2019-06-13
Accepted:	2019-06-14

Scientific Editor: Andrea Leibfried

Transaction Report:

February 12, 2019

Re: Life Science Alliance manuscript #LSA-2019-00305-T

Dr. Vidhya Madapusi Ravi
Uniklinikum Freiburg
Neurosurgery
Breisacher Str. 64
79106 Freiburg
Germany

Dear Dr. Ravi,

Thank you for submitting your manuscript entitled "Human Organotypic Brain Slice Culture: A novel framework for environmental research in NeuroOncology" to Life Science Alliance. The manuscript was assessed by expert reviewers, whose comments are appended to this letter.

As you will see, the reviewers appreciate the improved method provided and they provide constructive input on how to further strengthen your data and the representation of your work. The revision requests seem all straightforward to address, and we would thus like to invite you to submit a revised version of your manuscript to us.

Thank you for this interesting contribution to Life Science Alliance. We are looking forward to receiving your revised manuscript.

Sincerely,

B. MANUSCRIPT ORGANIZATION AND FORMATTING:

Reviewer #1 (Comments to the Authors (Required)):

Ravi et al. report on the use of human organotypic brain slices as a tool to study invasion of human glioblastoma cell lines. The topic is of high interest, the work on brain slices technically excellent and the outcome of importance. The work can be published, but I suggest a few modifications for better readability and quality.

1. Abstract: the abstract should be more precise and include also some raw data.
2. It is not fully clear why the authors mention data on rodents, as they do not show any data and the human experiments are a clear compact story. This work on rats is not well integrated.
3. It might be interesting for the reader to see a table on the details of all patients used for this study: e.g. sex, age, which type of tumor, death of patient, taken how long after diagnosis, size of tumor, MR data, etc.
4. what do the authors mean by mechanical dissociation (page 19), this is vague.
5. I am personally not happy to mix discussions with results, phrases like previous reports, this is in line, or suggest do not belong in results.
6. Rather I suggest to better discuss the data in context and make clearer subheadings; e.g. discuss viability of neurons, discuss serum-free, discuss limits and outlook, discuss in-growth-vascular, discuss brain slices, etc.
7. In this respect one issue might markedly improve the paper, the question on vascular innervation of the brain slices. Are new vessels formed, are the vessels altered, do vessels grow in or out? As the authors say astrocytes play a vital role in progression and angiogenesis of tumors.

Reviewer #2 (Comments to the Authors (Required)):

The authors describe the establishment of a protocol for organotypic slice culture from the human brain. The main point of this paper is that their model is suitable for neuro-oncology studies. The paper is well written and the data convincingly show what the authors claim. I did appreciate the effort the authors made in establishing a better standard for the slice culture procedure in a model (the human) that is the relevant one for clinical studies. I am therefore positive as for the publication of this paper, but I think the authors should definitely address the following points.

Here is my figure-by-figure comments to the authors.

Figure 2.

- (i) As for the cell loss during slice culture I do recommend the authors to add a TUNEL staining (or caspase) for the acute, 7d and 14d samples to evaluate the extent of apoptosis in the system. This simple staining will make the data set look more quantitative and controlled.
- (ii) Is lamination and overall tissue architecture affected by slice culture? The authors should address this point.

Figure 3.

- (i) Here I cite from the text: "We identified a significant loss of the cell-specific expression of neuronal genes between acute and cultured slices". It would be interesting for the readers if the authors were to dig further into the data in Figure 3a, and check if the loss of neuronal genes corresponds to a general loss of all genes (as a consequence of overall neurons loss), or to a loss of a specific set of genes (as consequence of a specific damage to neuronal polarity and/or architecture).

Although I understand that this might not be the focus of the authors work and research, I think adding this information will make the data interesting for a bigger audience, and that in turn will increase the impact of this work.

Figure 5.

(i) The data presented in this Figure are interesting, but I think the authors are under-using them. Here the authors missed the opportunity of provide a significant insight on the transcriptional differences in astrocytes extracted from two different type of tumors. For example, in panel c, it would be very important to run an unbiased analysis of the transcriptional profiles, focusing on the - potential- difference between astrocytes isolated from the mesenchimal and pro-neuronal GBM-induced tumor. This type of analysis could shed light on the cellular mechanisms responsible for the different infiltration behavior of the two cells lines.

Reviewer #3 (Comments to the Authors (Required)):

The work by Ravi et al. (LSA-2019-00305-T) "Human Organotypic Brain Slice Culture: A novel framework for environmental research in NeuroOncology" describes a human ex vivo model of slice cultures to study invasion in glioblastoma.

The topic is very important and interesting, since there is a substantial need for better models to mimic the interactions between glioblastoma cells and brain tissue. Authors have used a large battery of methods to describe the slice cultures and analysed the changes after glioblastoma cell injections.

However, because of the importance of this technique it was very disappointing to see that the methods section and the results were poorly represented. Both sections need to be thoroughly revised.

General remarks:

Please denote whether you are using standard deviation or standard error of the mean in your graphs and state the sample size for each experiment throughout the whole manuscript.

It is not clear whether the cell lines used originated from the same patients. Authors stressed the differences in immunological differences in xenograft models. Similar problems may play a role when co-cultures of human tissues of different origin are used. This aspect should be discussed.

The transcriptional characterization of the glioblastoma cell lines GCL-CL1 and GCL-CL2 is mentioned in the text. The parameters analysed are missing. They must be added.

Abstract lines 60-61: Please recheck the sentence.

Introduction:

Line 73: adapt the term glioblastoma multiforme to the new term of the recent WHO classification.

Lines 94-96: I agree with the authors that the presented model is superior to 2 dimensional models, but the scratch wound assay (described in the cited 2005 paper) is not state of the art anymore.

There are more complex/better models involving specific coatings or tumor/healthy cell derived extracellular matrix. This would be more appropriate to cite as state of the art.

Lines 96-101: The inclusion of subarchnoid space in this sentence should be reevaluated. In majority of cases glioblastomas extend in the brain parenchyma.

Lines 111-114: The citation (Tamimi and Juweid 2017) should be placed at right position. In their work there is no word on embryonic tissue.

Line 116: Please correct "preserves" to preserve

Results:

Lines 145-148: Please give arguments why long range interactions like chemical gradients or mechanical cues do not interfere with the tissue 2 cm away from the tumor boundary.

Lines 156-159: This data is not really shown.

Lines 161-163: $p > 0.05$ does not necessarily imply that there is no difference between two groups. If you want to provide such information use techniques similar to two one-sided test (TOST).

Line 166: Please also denote the significant differences between acute slices and 7/14 div slices in GFAP in figure 2d.

Line 176-178: Please explain how it might be possible that the expression levels of the analysed mRNAs are comparable when they obtained from different areas of the brain. Did the authors assume the same signature of occipital temporal or frontal cortices?

Lines 188-189: Please explain in detail about the positions of Recordings. Have the recordings been performed at the surface or in depth? How the authors standardize the process. Were the firing rates always similar for all areas?

Lines 193-194: Almost 20% of LDH level was measured after 14 div when compared to controls. Absolute values representing 100% must be added. The reason of decrease explained in detail.

Line 220/221: Why only a single time point (4d) of TMZ treatment? Why 250 μ M TMZ? This is approximately 50 μ g/ml being 25 fold higher than found in the liquor of patients (Ostermann 2004).

Line 239++ (Astrocyte Purification): Recent publications (Corbetta 2018) point out that GBM cells express GLAST. Could you provide a control that your cells are indeed astrocytes and not a mixed culture of astrocytes and glioblastoma cells?

Discussion:

Line 277: Add the missing bracket after "GBM".

Lines 282-284: Despite 2d culture models there are 3d culture models as well. Please discuss your model in the light of more recent in vitro models, e.g. 3d hyaluronic acid hydrogel models.

Lines 312-314: What is the MGMT status of the used glioblastoma cells? This might impact your results and discussion.

Lines 304-307: The authors state the invasion pattern to be similar to in vivo. Please explain the invasion pattern in more detail in the results section that allows such a conclusion.

Lines 325-328: The given values should be interpreted with more caution. In case of 40-50% astrocytes and 10-30% microglia the tumor mass should contain 20-50% tumor cells.

Methods:

Line 378: Please define "OR" on usage.

Line 392: Please specify the title. Are human or rat slices meant or both?

Line 398: It seems impossible to maximally culture the slices for 12 days but representing data at 14 div.

Lines 392-401: How many slices were obtained from one tissue block? Please add the range and the mean number.

Lines 403-416: Are the result based of one or of four slices?

Lines 424-425: Please state at least the method or a source of how the authors assessed the transduction quality.

Line 427: Please state the time of injection after slicing.

Lines 430-431: Slices are 300 μ m thick and shrink during culture time. How the thickness of slices changed during the culture period? Values at 1, 4, 7 and 14 div should be added. How did the authors assure to inject the tumor cells into a roughly similar depth and to avoid placing them on the membrane by penetrating the whole slice?

Line 449: "Injection was carried out 24h post slicing, to minimize trauma due to the sectioning procedure". How did the authors assessed that trauma is "low" after 24 h, since in rodent slices this needs approximately one week?

Line 455: Please clarify what sections are meant (rat, human, with our without GBM or all?).

Line 464: How was the threshold set? Which algorithm?

Line 464: Classical watershedding is efficient only if objects are roughly circular and if objects show no large overlap. Consequently, watershedding can lead to drastically wrong estimates of object counts. Please describe the procedure and results of your evaluation of the accuracy of object splitting.

Lines 471-478: What was the reason to use a 20x water immersion objective? The loss of quality is very high as visible in figure 4b.

Line 502: Please clarify the working principle of the applied MatLab script.

Line 521: Please define the quality score.

Line 524: Please define how data was post-processed.

Figures: The legends seem insufficient as the description in the text is not complete.

Figure 1: HE staining is too small to recognize structures. Furthermore, additional images from each state namely 1, 4, 7 and 14 div must be included in overview and in higher magnification.

Figure 2b: Please additionally show an overview image of the whole slice or a significantly larger area. Please make sure to show the isocortex layers to demonstrate structural preservation. Please improve the visibility of the scaling. In vitro cultures of the CNS regularly develop glial scar during the first days. Here no scar formation is visible and the number of GFAP positive cells is even lower in comparison to acute specimens. Please explain in detail the possible reasons for this phenomenon.

Figure 3a: Please show the following plots: acute vs 3 days; acute vs 7 days, 3 days vs 7 days to allow an accurate interpretation. A table should be included (e.g. as supplementary) showing the chosen signature genes and the percentage of up- or downregulated mRNA levels.

Figure 3b: Is there no significance between acute and 7div slices for microglia and neuron associated mRNA?

Figure 3c: Please insert a legend clarifying the colors.

Figure 4b: Please improve the visibility of the scaling.

Figure 4c: Please improve the display of the figure. In its current form it is hard to decipher.

Figure 6a: Why did the authors show the ELISA for one (?) cell line only? Is this pooled? Why not showing both separately, as they were stated to have a different molecular characterization (mesenchymal, proneural)?

Reviewer #1 (Comments to the Authors (Required)):

Ravi et al. report on the use of human organotypic brain slices as a tool to study invasion of human glioblastoma cell lines. The topic is of high interest, the work on brain slices technically excellent and the outcome of importance. The work can be published, but I suggest a few modifications for better readability and quality.

1. Abstract: the abstract should be more precise and include also some raw data.

We have now improved the abstract to contain a brief description of the primary results presented in this research report.

2. It is not fully clear why the authors mention data on rodents, as they do not show any data and the human experiments are a clear compact story. This work on rats is not well integrated.

The reviewer is right with this comment. We have now focused the manuscript with only data obtained from human patient donors and removed all extraneous data.

3. It might be interesting for the reader to see a table on the details of all patients used for this study: e.g. sex, age, which type of tumor, death of patient, taken how long after diagnosis, size of tumor, MR data, etc.

We have included a data table in the supplementary that contains the information regarding the patient donors, with details of patients age, gender, tissue type and time period between surgery and diagnosis. Since the samples were obtained from living patients undergoing therapeutic tissue resection, we are unable to provide information about the death of the patient donors.

4. What do the authors mean by mechanical dissociation (page 19), this is vague.

Line: 552: We have clarified the technique in the manuscript as mentioned previously by (Heinlein *et al*, 2010)

5. I am personally not happy to mix discussions with results, phrases like previous reports, this is in line, or suggest do not belong in results.

We would like to thank the reviewer for this comment. However, this style of reporting results with associated discussions is easier for the flow of the manuscript. A separate discussion section consolidating the results reported in the manuscript is also included after the results section.

6. Rather I suggest to better discuss the data in context and make clearer subheadings; e.g. discuss viability of neurons, discuss serum-free, discuss limits and outlook, discuss in-growth-vascular, discuss brain slices, etc.

The manuscript has been amended to reflect the request of the reviewer.

7. In this respect one issue might markedly improve the paper, the question on vascular innervation of the brain slices. Are new vessels formed, are the vessels altered, do vessels grow in or out? As the authors say astrocytes play a vital role in progression and angiogenesis of tumors.

As the reviewer requested, we performed additional experiments to stain and quantify the vascular innervation of our brain sections (Line 287 to 292). The results are added to Supplementary Fig 4. Since this does not directly fall within the scope of the manuscript, we have not added this information to the main manuscript.

Reviewer #2 (Comments to the Authors (Required)):

The authors describe the establishment of a protocol for organotypic slice culture from the human brain. The main point of this paper is that their model is suitable for neuro-oncology studies. The paper is well written and the data convincingly show what the authors claim. I did appreciate the effort the authors made in establishing a better standard for the slice culture procedure in a model (the human) that is the relevant one for clinical studies. I am therefore positive as for the publication of this paper, but I think the authors should definitely address the following points. Here is my figure-by-figure comments to the authors. Figure 2. (i) As for the cell loss during slice culture I do recommend the authors to add a TUNEL

staining (or caspase) for the acute, 7d and 14d samples to evaluate the extent of apoptosis in the system. This simple staining will make the data set look more quantitative and controlled.

We thank the reviewer for this comment since it does help improve the quality of the manuscript. We have added the TUNNEL assay data for DIV 1, 4, 7 &14 (Fig 3f,g) and text can be seen in Line 188 to 199.

(ii) Is lamination and overall tissue architecture affected by slice culture? The authors should address this point.

Based on our imaging data, the tissue cytoarchitecture remains relatively well preserved until DIV7. The data is included in Fig 3a.

Figure 3. (i) Here I cite from the text: "We identified a significant loss of the cell-specific expression of neuronal genes between acute and cultured slices". It would be interesting for the readers if the authors were to dig further into the data in Figure 3a, and check if the loss of neuronal genes corresponds to a general loss of all genes (as a consequence of overall neurons loss), or to a loss of a specific set of genes (as consequence of a specific damage to neuronal polarity and/or architecture). Although I understand that this might not be the focus of the authors work and research, I think adding this information will make the data interesting for a bigger audience, and that in turn will increase the impact of this work.

We have included more detailed information regarding the genetic expression which can be found in Fig 5 of the manuscript. As the reviewer has commented, further quantification is outside the scope of the manuscript. However, the data will be uploaded to GEO so that other researchers have the opportunity to further analyze the data to find correlations as mentioned by the reviewer.

Figure 5. (i) The data presented in this Figure are interesting, but I think the authors are under-using them. Here the authors missed the opportunity of provide a significant insight on the transcriptional differences in astrocytes extracted from two different type of tumors. For example, in panel c, it would be very important to run an unbiased analysis of the transcriptional profiles, focusing on the -potential- difference between astrocytes isolated from the mesenchimal and pro-neuronal GBM-induced tumor. This type of analysis could shed light on the cellular mechanisms

responsible for the different infiltration behavior of the two cells lines.

We agree with this point that the reviewer has raised. However, the primary focus of the work is to present a detailed method that can be used to maintain human neuronal sections viably for up to 14 days in culture, along with the generation of tumor microenvironment. We also present the possibility to viably extract specific cell types from this micro-environment to enable biological dissection of disease pathways. To demonstrate this, we made use of two validated cell lines with known properties and did a qPCR analysis on extracted astrocytes with a reduced gene set derived from previously published reports. The reviewers comment is highly valuable and will be considered for further work.

Reviewer #3 (Comments to the Authors (Required)):

The work by Ravi et al. (LSA-2019-00305-T) "Human Organotypic Brain Slice Culture: A novel framework for environmental research in NeuroOncology" describes a human ex vivo model of slice cultures to study invasion in glioblastoma. The topic is very important and interesting, since there is a substantial need for better models to mimic the interactions between glioblastoma cells and brain tissue. Authors have used a large battery of methods to describe the slice cultures and analysed the changes after glioblastoma cell injections. However, because of the importance of this technique it was very disappointing to see that the methods section and the results were poorly represented. Both sections need to be thoroughly revised. General remarks: Please denote whether you are using standard deviation or standard error of the mean in your graphs and state the sample size for each experiment throughout the whole manuscript.

We thank the reviewer for their constructive comment. We have tried to improve the manuscript to the best of our ability during this revision round. All figures are always denoted by standard deviation and we have updated the text to include the sample size.

It is not clear whether the cell lines used originated from the same patients. Authors stressed the differences in immunological differences in xenograft models. Similar problems may play a role when co-cultures of human tissues of different origin are used. This aspect should be discussed.

We strongly agree with the point that the reviewer has made. However, our justification to use cell lines sourced from another patient was so that we knew the properties of the cell lines that were being used, like the classification they fall under, RNA sequencing of the cell line and TMZ resistance so that we could validate the neuro-oncology research platform that we are presenting in this manuscript and that we can manipulate the environment and understand the circuits involved in the context of neuro-oncology.

The transcriptional characterization of the glioblastoma cell lines GCL-CL1 and GCL-CL2 is mentioned in the text. The parameters analysed are missing. They must be added.

This information has been previously published by our research group and have been cited in the manuscript: Line 245

Abstract lines 60-61: Please recheck the sentence.

The sentence has been amended to improve readability.

Introduction: Line 73: adapt the term glioblastoma multiforme to the new term of the recent WHO classification.

Line:76: The text has been updated.

Lines 94-96: I agree with the authors that the presented model is superior to 2 dimensional models, but the scratch wound assay (described in the cited 2005 paper) is not state of the art anymore. There are more complex/better models involving specific coatings or tumor/healthy cell derived extracellular matrix. This would be more appropriate to cite as state of the art.

The manuscript has been updated to include this information this can be found from line 103 to 109

Lines 96-101: The inclusion of subarchnoid space in this sentence should be reevaluated. In majority of cases glioblastomas extend in the brain parenchyma.

Line:111: The text has been updated as per the reviewer's recommendation.

Lines 111-114: The citation (Tamimi and Juweid 2017) should be placed at right position. In their work there is no word on embryonic tissue.

The text has been revised and citation corrected.

Line 116: Please correct "preserves" to preserve

Line:117: The text has been updated.

Results:

Lines 145-148: Please give arguments why long range interactions like chemical gradients or mechanical cues do not interfere with the tissue 2 cm away from the tumor boundary.

We thank the reviewer for raising this interesting point. There is no guarantee that there is no interaction between the tissue used in this study and the infiltrating front of the tumor. However, this is the healthiest tissue that can be obtained from a living patient.

Lines 156-159: This data is not really shown.

The manuscript has been updated to include this information (Fig 3a,b,c,d,e or line 159 to 187).

Lines 161-163: $p > 0.05$ does not necessarily imply that there is no difference between two groups. If you want to provide such information use techniques similar to two one-sided test (TOST).

We have performed additional experiments to further validate the results that were presented. With the increased sample size, we report that there is a decrease in the number of neurons over the culture period, however, there is no change in the rate at which the neurons are lost over the entire culture duration. The tissue sections retain 80% of the total number of neurons in comparison to acute sections, which is now presented in the updated manuscript (Fig 3b,c,d,e or line 163 to 187)

Line 166: Please also denote the significant differences between acute slices and 7/14div slices in GFAP in figure 2d.

The figures have been updated to reflect this information. This can be found in Fig 3 d,e.

Line 176-178: Please explain how it might be possible that the expression levels of the analysed mRNAs are comparable when they obtained from different areas of the brain. Did the authors assume the same signature of occipital temporal or frontal cortices?

We thank the reviewer for this highly insightful comment. We missed mentioning that the RNA sequencing was carried out only from samples obtained from the temporal cortex. The manuscript has been amended to include this information (Fig 5a or line 217 o 219).

Lines 188-189: Please explain in detail about the positions of Recordings. Have the recordings been performed at the surface or in depth? How the authors standardize the process. Were the firing rates always similar for all areas?

The manuscript has been updated to include all the information regarding the electrophysiological recordings. (Fig 4a,b,c or Line 207 to 216).

Lines 193-194: Almost 20% of LDH level was measured after 14 div when compared to controls. Absolute values representing 100% must be added. The reason of decrease explained in detail.

Experiments were repeated again along with TUNNEL assay and data were represented in 100% absolute values. Since the LDH value was measured by means of ELISA, we only have an absorbance value which, to our understanding, by itself does not provide any value to the reader. (Fig 3f,g,h,i or Line 188 to 206)

Line 220/221: Why only a single time point (4d) of TMZ treatment? Why 250 μ M TMZ? This is approximately 50 μ g/ml being 25 fold higher than found in the liquor of patients (Ostermann 2004).

We thank the reviewer for pointing out obvious errors in the methodology. We performed additional experiments that reflect physiological conditions and the manuscript has been updated with this information (Fig 6b or Line 261 to 275).

Line 239++ (Astrocyte Purification): Recent publications (Corbetta 2018) point out that GBM cells express GLAST. Could you provide a control that your cells are indeed astrocytes and not a mixed culture of astrocytes and glioblastoma cells?

We have performed additional experiments where we report that there is minimal contamination of the extracted astrocytes with GBM cells. To validate the non-contaminated status of the astrocytes post MACS separation, we performed FACS and report <1% presence of ZsGreen tumor cells in the samples purified using GLAST antibody. The manuscript has been updated to reflect this information. (Supplementary Fig 5 or line 308 to 311)

Discussion:

Line 277: Add the missing bracket after "GBM".

The manuscript has been updated.

Lines 282-284: Despite 2d culture models there are 3d culture models as well. Please discuss your model in the light of more recent in vitro models, e.g. 3d hyaluronic acid hydrogel models.

The manuscript has been updated to include all the information regarding other 3d models (line 340 to 342).

Lines 312-314: What is the MGMT status of the used glioblastoma cells? This might impact your results and discussion.

MGMT status of the cell lines is mentioned in the results and discussion part. The proneural cell type GSC_CL1 is MGMT methylated while the mesenchymal GSC_CL2 is MGMT non-methylated. The status is clearly visible in the results that show the effect that TMZ has on the proliferation profile of the utilized cell lines.(Line 378 to 380)

Lines 304-307: The authors state the invasion pattern to be similar to in vivo. Please explain the invasion pattern in more detail in the results section that allows such a conclusion.

The manuscript has been updated to include the appropriate references describing this phenomenon and figures have been included to highlight this fact (Line 369 to 372, Line 380 to 383).

Lines 325-328: The given values should be interpreted with more caution. In case of 40-50% astrocytes and 10-30% microglia the tumor mass should contain 20-50% tumor cells.

The manuscript has been updated with corrected values according to previous reports (Zhang *et al*, 2016) and (Placone *et al*, 2016). (Line 390- 393).

The text Methods:

Line 378: Please define "OR" on usage.

We have defined the term OR. Just for clarification, it refers to the 'Operating Room'

Line 392: Please specify the title. Are human or rat slices meant or both?

The first version of the manuscript contained both. The updated manuscript reports on work done only in human donor samples.

Line 398: It seems impossible to maximally culture the slices for 12 days but representing data at 14 div.

We thank the reviewer for pointing out this glaring error. We have updated the text to reflect actual numbers.

Lines 392-401: How many slices were obtained from one tissue block? Please add the range and the mean number.

The manuscript has been updated to include this information (Fig 2b or Line 465 to 467).

Lines 403-416: Are the result based of one or of four slices?

The results were based on four slices since the biomass of 1 single slice is too low for any quantification. 4 slices give us about 500mg of tissue sample.

Lines 424-425: Please state at least the method or a source of how the authors assessed the transduction quality.

The transduction quality was measured by regular microscope images.

Line 427: Please state the time of injection after slicing.

The manuscript has been updated to contain this information. Injection was carried out 24 hours post resection and sectioning (Fig 6a or line 241 to 245).

Lines 430-431: Slices are 300 μ m thick and shrink during culture time. How the thickness of slices changed during the culture period? Values at 1, 4, 7 and 14 div should be added. How did the authors assure to inject the tumor cells into a roughly similar depth and to avoid placing them on the membrane by penetrating the whole slice?

We thank the reviewer for this insightful comment. Indeed, there are previous reports regarding the shrinkage of slices obtained from rodent and human slices. To validate these results, we made use of a live imaging system to image the slices over the entire duration of the culture period. To our surprise, the slices show no change in the area during the culture period (up to DIV10). Since these results significantly contradict all previously published reports, we decided to not present this data since further validation needs to be carried out. (Please check the file Section Flattening.png)

The injection was carried out by means of a modified stereotactic apparatus, under microscopic control. Briefly, the tip of the injection needle was positioned to create a dimple on the surface of the section. The tip was then lowered till the surface was breached and the cells were injected. Since the cells are tagged with zsGreen, microscopic validation could be carried out regarding the position of the injected cells.

Line 449: "Injection was carried out 24h post slicing, to minimize trauma due to the sectioning procedure". How did the authors assessed that trauma is "low" after 24 h, since in rodent slices this needs approximately one week?

The results from the TUNEL, LDH assay and GFAP⁺ astrocytes show a significantly lower value on DIV1 in comparison to DIV0 (Fig 3 f, g, h, i). To our understanding, the injection of GBM cells into the slice post traumatic resection and sectioning might be too much for the tissue to recover from. Therefore, the 24 hours post sectioning time point was chosen for GBM injection.

Line 455: Please clarify what sections are meant (rat, human, with or without GBM or all?).

The manuscript has been updated to improve clarity (Line 463).

Line 464: How was the threshold set? Which algorithm?

Line 464: Classical watershedding is efficient only if objects are roughly circular and if objects show no large overlap. Consequently, watershedding can lead to drastically wrong estimates of object counts. Please describe the procedure and results of your evaluation of the accuracy of object splitting.

To avoid any errors in cell count estimation, we made use of a blinded expert for the counting and compared the values obtained by watershedding based automatic counting. We report a <10% difference in the values obtained by the algorithm vs a human expert and therefore use these values in the manuscript (Fig 3b,c).

Lines 471-478: What was the reason to use a 20x water immersion objective? The loss of quality is very high as visible in figure 4b.

The only reason for using the 20x objective was that it is the the only available lens on the 2-Photon setup used in this study.

Line 502: Please clarify the working principle of the applied MatLab script.

The manuscript has been updated to include this information. The detailed information can be found in a previously reported study that has been cited in the text (Line 597).

Line 521: Please define the quality score.

The quality score is measured based on the base calling algorithm albacore (nanopore), defined as $-10 \cdot \log_{10}(\text{Probability of incorrect base call})$.

Line 524: Please define how data was post-processed.

The data are processed by the online available script: <https://github.com/heilandd/NanoPoreSeq>

Figures: The legends seem insufficient as the description in the text is not complete.

The manuscript has been updated with updated and improved figure legends.

Figure 1: HE staining is too small to recognize structures. Furthermore, additional images from each state namely 1, 4, 7 and 14 div must be included in overview and in higher magnification.

The manuscript has been updated with new figure panels to improve visibility (Fig 2a).

Figure 2b: Please additionally show an overview image of the whole slice or a significantly larger area. Please make sure to show the isocortex layers to demonstrate structural preservation. Please improve the visibility of the scaling.

The manuscript has been updated with new figure panels showing the structural preservation (Fig 3a) with all information regarding scale bars in the legends.

In vitro cultures of the CNS regularly develop glial scar during the first days. Here no scar formation is visible and the number of GFAP positive cells is even lower in comparison to acute specimens. Please explain in detail the possible reasons for this phenomenon.

The manuscript has been updated to include our reasoning for this phenomenon (line 186 to 187).

Figure 3a: Please show the following plots: acute vs 3 days; acute vs 7 days, 3 days vs 7 days to allow an accurate interpretation. A table should be included (e.g. as supplementary) showing the chosen signature genes and the percentage of up- or downregulated mRNA levels.

The manuscript has been updated with detailed figures with respect to the data from the RNAseq results. All the data will be uploaded to GEO so that other researchers have the opportunity to further analyze the data to find correlations as mentioned by the reviewer.

Figure 3b: Is there no significance between acute and 7div slices for microglia and neuron associated mRNA?

The manuscript has been updated with detailed figures with respect to the data from the RNAseq results. All the data will be uploaded to GEO so that other researchers have the opportunity to further analyze the data to find correlations as mentioned by the reviewer.

Figure 3c: Please insert a legend clarifying the colors.

The manuscript has been updated with new figure panels to improve visibility (Fig 4b).

Figure 4b: Please improve the visibility of the scaling.

The manuscript has been updated with new figure panels to improve visibility (Fig 6e).

Figure 4c: Please improve the display of the figure. In its current form it is hard to decipher.

The manuscript has been updated with new figure panels to improve visibility (Fig 6d).

Figure 6a: Why did the authors show the ELISA for one (?) cell line only? Is this pooled? Why not showing both separately, as they were stated to have a different molecular characterization (mesenchymal, proneural)?

The ELISA based cytokine measurements have been individually represented with a new figure panel included with this information (Fig 7).

Bibliography

- Heinlein C, Deppert W, Braithwaite AW & Speidel D (2010) A rapid and optimization-free procedure allows the in vivo detection of subtle cell cycle and ploidy alterations in tissues by flow cytometry. *Cell Cycle* **9**: 3584–3590
- Placone AL, Quiñones-Hinojosa A & Searson PC (2016) The role of astrocytes in the progression of brain cancer: complicating the picture of the tumor microenvironment. *Tumour Biol.* **37**: 61–69
- Zhang Y, Sloan SA, Clarke LE, Caneda C, Plaza CA, Blumenthal PD, Vogel H, Steinberg GK, Edwards MSB, Li G, Duncan JA, Cheshier SH, Shuer LM, Chang EF, Grant GA, Gephart MGH & Barres BA (2016) Purification and Characterization of Progenitor and Mature Human Astrocytes Reveals Transcriptional and Functional Differences with Mouse. *Neuron* **89**: 37–53

Normalized area

2.0
1.5
1.0
0.5

0 1 2 3 4 5 6 7 8 9 10

Time (Days)

June 6, 2019

RE: Life Science Alliance Manuscript #LSA-2019-00305-T R

Dear Dr. Ravi,

Thank you for submitting your revised manuscript entitled "Human Organotypic Brain Slice Culture: A novel framework for environmental research in NeuroOncology".

As you will see, the reviewers appreciate the introduced changes and reviewer #3 provides constructive input on how to further improve your manuscript. We would thus be happy to accept your manuscript for publication in Life Science Alliance, pending minor revision to address the the remaining concerns of reviewer #3 and to adhere to our formatting guidelines:

- please submit the manuscript text as a docx file
- please mention Fig3H and I in the legend
- please mention Fig4D-G in the manuscript text and in the legend
- please mention FigS2A and B in the legend
- please deposit and provide accession codes for the RNA-seq data
- please indicate in the figure legends which statistical test has been used
- please enter all authors and author contributions in our submission system
- please add a summary blurb for your work

A. FINAL FILES:

-- Summary blurb (enter in submission system): A short text summarizing in a single sentence the study (max. 200 characters including spaces). This text is used in conjunction with the titles of

papers, hence should be informative and complementary to the title. It should describe the context and significance of the findings for a general readership; it should be written in the present tense and refer to the work in the third person. Author names should not be mentioned.

B. MANUSCRIPT ORGANIZATION AND FORMATTING:

Sincerely,

Andrea Leibfried, PhD
Executive Editor
Life Science Alliance
Meyerohofstr. 1
69117 Heidelberg, Germany
t +49 6221 8891 502
e a.leibfried@life-science-alliance.org
www.life-science-alliance.org

Reviewer #2 (Comments to the Authors (Required)):

In the revision, the authors addressed the points raised by the referees in a satisfactory manner. I think that the paper, in its present revised form, is ready to be accepted for publication.

Reviewer #3 (Comments to the Authors (Required)):

In the revised version the authors have performed additional experiments and addressed almost all points raised by this reviewer. The manuscript is substantially improved and I would like to thank the authors for their efforts. Below mentioned there are few minor points remained that need further consideration before final acceptance.

Abstract:

Line 61: replace significance by significant

Lines 63, 143, 161 and others: recheck and remove capitalizations. glioblastoma, epilepsy immunohistochemistry etc.

Results:

Line 144: According to the supplemental table there were at least two significantly younger donors. Please correct the sentence accordingly. Or does the description only refers to glioblastoma patients? If so please clarify.

Lines 152-154 please rephrase this sentence.

Lines 213/214: Figure 3b shows measurements for 1 div and 14 div (according to its labeling) rather than acute and 14 div. Please correct.

Lines 214/215: The extracellular spikes seem to be more frequent in 14 div samples in figure 4c. Is this a general phenomenon?

Lines 249-252: It is not clear what the authors are referring to looking at their time frames. Tumor cells were injected 1 day after preparation (lines 240/241), and imaged 4, 7 and 14 days after preparation, as well as directly after tumor injection. Looking at the respective figure I suppose that the authors are referring to 4 days after preparation? If so I see no "cloudy" invasion pattern for GSC-CL1. Please clarify.

Lines 316-322: Please refer to A(n) or A(2) as the non-inflammatory subtype and do not use both terms.

Discussion:

Lines 360-361: Please rephrase, because significantly different expression was observed between acute and cultured sections (supplemental figure 1b , neuron), even though differences were small.

Methods:

Line 437: "at least"

Line 472: "three to four"

Line 509 (and others): Centrifugation: 5000g (no capital "g")

Lines 512, 515: μL or μL

The methods section is missing of how "intersections" were measured and defined (supplemental figure 4b), but only states how vessels were detected.

Figures:

Figure 1b: Please correct/cut off the graph for the density estimation, as there is no age below zero.

Figure 5b: Please indicate the cut-off values used for determining low and high gene expression.

Figure 5d-g: Please use an appropriate scale for the y-axis.

Supplementary figure 5a: The gating seems off. Why cutting through the peak with the center at SSC around 10^4 ? I would suspect these to be cells as well. If so would this significantly change the astrocyte to tumor cell ratio?

Supplemental Videos 1/2: It is not clear what is shown.

Supplementary Videos 5/6: They do not seem to work. Please check this.

General remark to supplemental videos: A short explanation is needed in similarity to the figures.

Reviewer #2 (Comments to the Authors (Required)):

In the revision, the authors addressed the points raised by the referees in a satisfactory manner. I think that the paper, in its present revised form, is ready to be accepted for publication.

We would like to thank the reviewer for his time and comments leading to an improvement of the manuscript.

Reviewer #3 (Comments to the Authors (Required)):

In the revised version the authors have performed additional experiments and addressed almost all points raised by this reviewer. The manuscript is substantially improved and I would like to thank the authors for their efforts. Below mentioned there are few minor points remained that need further consideration before final acceptance.

We would like to thank the reviewer for his time and comments leading to an improvement of the manuscript.

Abstract:

Line 61: replace significance by significant

Lines 63, 143, 161 and others: recheck and remove capitalizations. glioblastoma, epilepsy immunohistochemistry etc.

The text has been improved to include the changes requested.

Results:

Line 144: According to the supplemental table there were at least two significantly younger donors. Please correct the sentence accordingly. Or does the description only refers to glioblastoma patients? If so please clarify.

Line 148-150: The 2 donor categories have been mentioned separately with their specific age ranges.

Lines 152-154 please rephrase this sentence.

The text has been improved to ensure clarity.

Lines 213/214: Figure 3b shows measurements for 1 div and 14 div (according to its labeling) rather than acute and 14 div. Please correct.

Thanks for pointing out this mistake. It is Fig 4b and not 3b. And the text has been changed accordingly.

Lines 214/215: The extracellular spikes seem to be more frequent in 14 div samples in figure 4c. Is this a general phenomenon?

The displayed segment is a random sampling of the recorded signal. We do not see a significant difference between the measured time points.

Lines 249-252: It is not clear what the authors are referring to looking at their time frames. Tumor cells were injected 1 day after preparation (lines 240/241), and imaged 4, 7 and 14 days after preparation, as well as directly after tumor injection. Looking at the respective figure I suppose that the authors are referring to 4 days after preparation? If so I see no "cloudy" invasion pattern for GSC-CL1. Please clarify.

We thank the reviewer for picking up on this lack of clarity in presentation. We have improved the text and figure legend for clarity. The tumor cells were injected into the slice one day post plating, which is DIV 1. The time points post injection have been now denoted as days post injection (DPI). In this work, we have imaged immediately after injection which is DPI 0 followed by DPI 4, DPI 7 and DPI 14. In the initial two days (which is DPI 1,2) there was a cloud like pattern which is not shown here.

Lines 316-322: Please refer to A(n) or A(2) as the non-inflammatory subtype and do not use both terms.

Line 318: The text has been changed.

Discussion:

Lines 360-361: Please rephrase, because significantly different expression was observed between acute and cultured sections (supplemental figure 1b, neuron), even though differences were small.

Line 360 to 364: The sentence has been rephrased to improve clarity.

Methods:

Line 437: "at least"

Changed.

Line 472: "three to four"

Changed.

Line 509 (and others): Centrifugation: 5000g (no capital "g"), Lines 512, 515: μ l or μ L

Changed.

The methods section is missing of how "intersections" were measured and defined (supplemental figure 4b), but only states how vessels were detected.

Line 549-551: Intersections were measured by means of SHOLL analysis. The text has been updated to include this information.

Figures:

Figure 1b: Please correct/cut off the graph for the density estimation, as there is no age below zero.

Changed.

Figure 5b: Please indicate the cut-off values used for determining low and high gene expression.

All genes were used in the case of this analysis.

Figure 5d-g: Please use an appropriate scale for the y-axis.

The global color scale is depicted on the top right-hand side of Fig 5. The legend has been updated to include this information for clarity to the audience.

Supplementary figure 5a: The gating seems off. Why cutting through the peak with the center at SSC around 10^4 ? I would suspect these to be cells as well. If so would this significantly change the astrocyte to tumor cell ratio?

Thanks for pointing this out. We have readjusted the gating to include the whole cluster, and the tumor contamination has been reduced from 0.07% to 0.04%.

Supplemental Videos 1/2: It is not clear what is shown.

Legends are included.

Supplemental Videos 5/6: They do not seem to work. Please check this.

New files have been generated and reuploaded.

General remark to supplemental videos: A short explanation is needed in similarity to the figures.

The legend has been updated with more information.

June 14, 2019

RE: Life Science Alliance Manuscript #LSA-2019-00305-TRR

Dear Dr. Ravi,

Thank you for submitting your Research Article entitled "Human Organotypic Brain Slice Culture: A novel framework for environmental research in NeuroOncology". We appreciate the introduced changes and it is a pleasure to let you know that your manuscript is now accepted for publication in Life Science Alliance. Congratulations on this interesting work.

DISTRIBUTION OF MATERIALS:

Again, congratulations on a very nice paper. I hope you found the review process to be constructive and are pleased with how the manuscript was handled editorially. We look forward to future exciting submissions from your lab.

Sincerely,

Andrea Leibfried, PhD
Executive Editor
Life Science Alliance
Meyerhofstr. 1

69117 Heidelberg, Germany
t +49 6221 8891 502
e a.leibfried@life-science-alliance.org
www.life-science-alliance.org